# LogicBench: A Benchmark for Evaluation of Logical Reasoning

## Abstract

Recently developed large language models (LLMs) have been shown to perform
remarkably well on a wide range of language understanding tasks. But, can they
really "Reason" over the natural language? This question has been receiving signif-
icant research attention and a number of reasoning skills such as commonsense,
numerical, and qualitative have been studied. However, the crucial skill pertaining
to 'logical reasoning' has remained underexplored. Existing work investigating
this reasoning ability has focused only on a couple of axioms (such as modus
ponens and modus tollens) of propositional and first-order logic. To study logical
reasoning, we introduce *LogicBench*, a systematically created natural language
question-answering dataset encompassing 25 reasoning patterns spanning over
propositional, first-order, and non-monotonic logics. Key steps of our dataset
construction consist of (1) controlled generation of sentences and their negations
containing different ontologies, (2) *(context, question, answer)* triplets creation us-
ing heuristically designed templates, and (3) semantic variations of triplets adding
more diversity. We first evaluate easily accessible and widely used LLMs such as
GPT-3, ChatGPT, and FLAN-T5 and show that they do not fare well on *LogicBench*,
achieving just above random accuracy on average ($\sim 52\%$). Then, we show that
LLMs trained using our data exhibit a better understanding of logical reasoning
leading to performance improvements on several existing logical reasoning datasets
such as LogicNLI, FOLIO, LogiQA, and ReClor.[1]

## 1  Introduction

Large language models such as GPT-3 [3], ChatGPT, and FLAN [18] have made remarkable progress
in NLP research enabling machines to perform a variety of language tasks that were previously
thought to be exclusive to humans [12, 2, 20]. However, the ability of these LLMs to reason
"logically" over natural language text remains under-explored, even though logical reasoning is a
fundamental aspect of intelligence and a crucial requirement for many practical applications, such
as question-answering systems [8] and conversational agents [1]. Although several datasets have
been proposed [4, 16, 7, 13] to evaluate the logical reasoning capabilities of LLMs, these datasets
are limited in their scope by (1) not evaluating logical reasoning independently of other forms of
reasoning such as LogiQA [11] and ReClor [19]; and (2) evaluating only a single type of logic and
covering only few logical inference rules as done in FOLIO [6] and ProntoQA [14]. Thus, our aim in
this work is to address the lacuna of having a more comprehensive evaluation dataset for LLMs.

To this end, we propose *LogicBench*, a systematically created question-answering dataset for the
evaluation of logical reasoning ability. As illustrated in Figure 1, *LogicBench* includes a total of 25

---

[1]Data is available at `https://anonymous.4open.science/r/LogicBench-EEBB`

Submitted to the 37th Conference on Neural Information Processing Systems (NeurIPS 2023) Track on Datasets
and Benchmarks. Do not distribute.

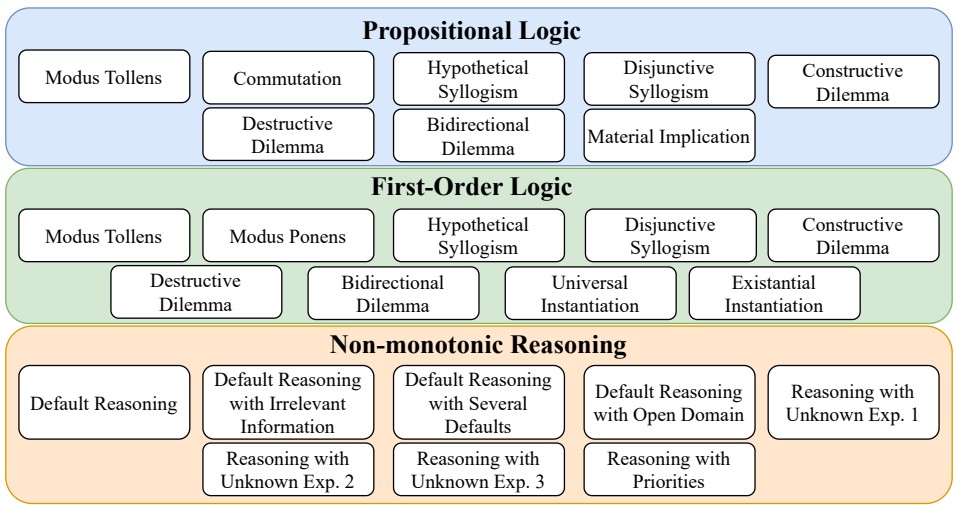

Figure 1: Comprehensive representation of different inference rules and reasoning patterns covered by propositional, first-order, and non-monotonic logics. *Exp.* indicates Expectation

reasoning patterns across propositional, first-order, and non-monotonic logics. To evaluate LLMs, we formulate a binary classification task in *LogicBench* in which the context represents logical statements and the models have to determine whether a conclusion given in the question is logically entailed by the context. For example, given the context "All mammals have fur" and "A cat is a mammal", for the question is "Does a cat have fur?", the correct answer, is "Yes". (Additional examples of task instances are presented in Table 3 and Appendix B. To construct *LogicBench*, we use a three-stage procedure (refer to §2). In the first stage, we prompt GPT-3 to generate a variety of coherent natural language sentences consisting of different 'ontologies' (i.e., a collection of concepts such as car, person, and animals) and their corresponding negations (refer to §2.2.1). Then, in the second stage, we generate *(context, question, answer)* triplets using heuristically designed templates based on the inference rules and patterns. Finally, in the third stage, we generate semantics preserving and inverting variations of these logical rules by incorporating negations.

We evaluate a range of accessible and widely used LLMs including GPT-3 [3], ChatGPT, FLAN-T5 [18], Tk-instruct [17], and UnifiedQA [9] with respect to *LogicBench* on the accuracy of the predicted answers (i.e., "Yes" or "No"). Experimental results reveal that these models struggle with respect to many of the inference rules and patterns (showing $\sim 52\%$ accuracy on an average), suggesting significant room for improvement in their logical reasoning abilities. We then synthetically augment LogicBench and train T5-large. Our initial experimental results show that this improves the logical reasoning ability of existing models leading to performance improvement on other logic datasets, such as LogicNLI, and FOLIO ($\sim 2\%$ on an average), and shows competitive performance on LogiQA and ReClor. In summary, our contributions are as follows:

1. Introducing *LogicBench*: A systematically created dataset to assess the logical reasoning capabilities of LLMs across propositional, first-order, and non-monotonic logics. This benchmark will be publicly available for evaluation and training purposes.
2. We propose a three-stage method to construct *LogicBench* consisting of GPT-3 to generate coherent natural language sentences using prompts and a template-based module to convert them into logical rules. By assessing the performance of existing LLMs, we gain insights into their logical reasoning abilities which further leads to several interesting findings.
3. To the best of the authors' knowledge, this is the first benchmark to study non-monotonic reasoning, as well as various inference rules in propositional and first-order logics including hypothetical and disjunctive syllogism; and bidirectional, constructive, and destructive dilemmas in the NLP domain.

## 2 LogicBench

In this section we discuss the logic types, inference rules, and patterns that are explored in this research. We also outline the methods for generating the data, and statistics of *LogicBench*.

### 2.1 Logics Types

**Propositional Logic (PL)**  Propositional logic employs a collection of statements or propositions (denoted as $\mathcal{P} = p_1, p_2, ..., p_n$, where $p_i$ represents a proposition) and builds upon them using logical connectives such as '$\wedge$', '$\vee$', '$\rightarrow$', '$\leftrightarrow$', and '$\neg$'. Several inference rules for propositional logic have been defined using which given a set of premises, one can derive a sound conclusion. To illustrate this, let us consider two propositions: $p_1$, which states "It is raining," and $p_2$, which states "It is cloudy." From these propositions, we can construct a context (KB) consisting of two premises: (1) $p_1 \rightarrow p_2$ and (2) $p_1$. Based on this KB, we can conclude $p_2$. This inference rule is written as $((p_1 \rightarrow p_2) \wedge p_1) \vdash p_2$ and is known as 'Modus Ponens'. In our study, we explore nine distinct inference rules of propositional logic, extensions of seven of them with one-variable and a universal quantifier, and two axioms of first-order logic as shown in Table 1. These inference rules provide a systematic framework for deriving valid conclusions.

| Names | Propositional Logic | Extension to a (restricted) First-order Logic |
|---|---|---|
| MP | $((p \rightarrow q) \wedge p) \vdash q$ | $(\forall x(p(x) \rightarrow q(x)) \wedge p(a)) \vdash q(a)$ |
| MT | $((p \rightarrow q) \wedge \neg q) \vdash \neg p$ | $(\forall x(p(x) \rightarrow q(x)) \wedge \neg q(a)) \vdash \neg p(a)$ |
| HS | $((p \rightarrow q)) \wedge (q \rightarrow r)) \vdash (p \rightarrow r)$ | $(\forall x((p(x) \rightarrow q(x)) \wedge (q(x) \rightarrow r(x))) \vdash (p(a) \rightarrow r(a))$ |
| DS | $((p \vee q) \wedge \neg p) \vdash q$ | $(\forall x(p(x) \vee q(x)) \wedge \neg p(a)) \vdash q(a)$ |
| CD | $((p \rightarrow q) \wedge (r \rightarrow s) \wedge (p \vee r)) \vdash (q \vee s)$ | $(\forall x((p(x) \rightarrow q(x)) \wedge (r(x) \rightarrow s(x))) \wedge (p(a) \vee r(a))) \vdash (q(a) \vee s(a))$ |
| DD | $((p \rightarrow q) \wedge (r \rightarrow s) \wedge (\neg q \vee \neg s)) \vdash (\neg p \vee \neg r)$ | $(\forall x((p(x) \rightarrow q(x)) \wedge (r(x) \rightarrow s(x))) \wedge (\neg q(a) \vee \neg s(a))) \vdash (\neg p(a) \vee \neg r(a))$ |
| BD | $((p \rightarrow q) \wedge (r \rightarrow s) \wedge (p \vee \neg s)) \vdash (q \vee \neg r)$ | $(\forall x((p(x) \rightarrow q(x)) \wedge (r(x) \rightarrow s(x))) \wedge (p(a) \vee \neg s(a))) \vdash (q(a) \vee \neg r(a))$ |
| CT | $(p \vee q) \vdash (q \vee p)$ | - |
| MI | $(p \rightarrow q) \vdash (\neg p \vee q)$ | - |
| EI | - | $\exists x P(x) \Rightarrow P(a)$ |
| UI | - | $\forall x A \Rightarrow A\{x \mapsto a\}$ |

Table 1: Inference rules and (two) axioms that establish the relationship between premises and their corresponding conclusions. MP: Modus Ponens, MT: Modus Tollens, HS: Hypothetical Syllogism, DS: Disjunctive Syllogism, CD: Constructive Dilemma, DD: Destructive Dilemma, BD: Bidirectional Dilemma, CT: Commutation, MI: Material Implication, EI: Existential Instantiation, UI: Universal Instantiation

**First-order Logic (FOL)**  In this work, we consider a restricted set of logical axioms for FOL that utilize quantifiers, $\forall$ (universal quantifier) and $\exists$ (existential quantifier). The universal quantifier ($\forall$) denotes that a statement holds true for all instances within a specific category. In contrast, the existential quantifier ($\exists$) indicates that a statement is true for at least one instance within its scope. For instance, a simple extension of propositional 'Modus Ponens' is an inference rule where given the premises $\forall(p(x) \rightarrow q(x))$ and $p(a)$, we conclude $q(a)$ (e.g., given "All kings are greedy" and "Sam is a king", we can conclude "Sam is greedy"). Here, we explore various axioms and inference rules that incorporate the quantifiers shown in Table 1.

**Non-monotonic (NM) Reasoning**  In this work, we analyze a range of logical reasoning templates in NM logics involving "Default Reasoning," "Reasoning about Unknown Expectations," and "Reasoning about Priorities." These templates are inspired by the compilation [10] made in 1989 to evaluate the abilities of various non-monotonic logics that were being developed at that time. Below Table 2 shows examples of NM reasoning. Additional examples are given in Appendix B.3.

A key aspect of NM logics is to formalize notions such as "normally," "typically," and "usually" that are not directly formalizable using classical quantifiers in the first-order setting. The general rule "Heavy blocks are normally located on the table" does not imply that "All heavy blocks are

| Basic Default Reasoning | Default Reasoning with Irrelevant Information |
|---|---|
| Context: Blocks A and B are heavy. Heavy blocks are typically located on the table. A is not on the table. 

 Conclusion: B is on the table. | Context: Blocks A and B are heavy. Heavy blocks are typically located on the table. A is not on the table. B is red. 

 Conclusion: B is on the table. |
| **Reasoning about Unknown Expectations** | **Reasoning about Priorities** |
| Context: Blocks A, B, and C are heavy. Heavy blocks are normally located on the table. At least one of A, B is not on the table. 

 Conclusion: C is on the table. Exactly one of A, B is not on the table. | Context: Jack asserts that block A is on the table. Mary asserts that block A is not on the table. When people assert something, they are normally right. 

 Conclusion: If Mary's evidence is more reliable than Jack's. then block A is not on the table |

Table 2: Illustrative examples of non-monotonic reasoning adapted from [10]

always located on the table". Rather, this rule allows for exceptions. Our work explores various NM reasoning types, as depicted in Figure 1, to delve deeper into the nuances of this type of reasoning.

## 2.2 Data Creation

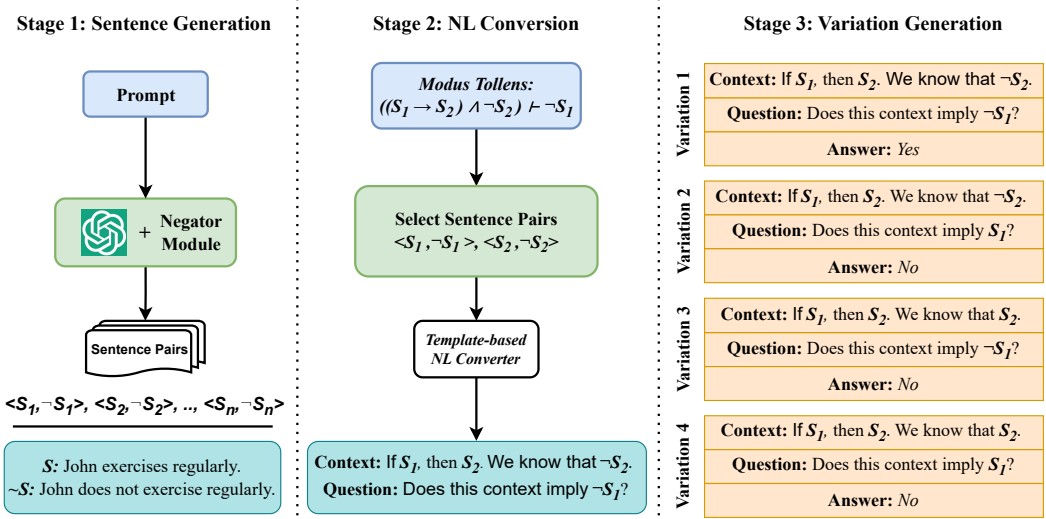

Figure 2: Schematic representation of three-stage procedure for data creation. NL: Natural Language

Our data creation procedure, illustrated in Figure 2, consists of three stages:

1. **Sentence Generation:** Starting with a given prompt, we generate coherent sentences and their negations that incorporate different ontologies.
2. **NL Conversion:** Using predefined templates of reasoning patterns based on their formal expressions, we convert the generated sentences into *(context, question, answer)* triplets.
3. **Variation Generation:** We generate semantically preserving and inverting variations of these triplets to add more diversity.

By following this method, we construct *LogicBench*, and examples of generated data corresponding to each logic type and reasoning patterns are presented in Appendix B.

### 2.2.1 Sentence Generation

Here, the first step is to generate sentences with diverse *ontologies*. An ontology represents a collection of concepts (e.g. car, person, animals, etc.) along with their corresponding associated

properties. To generate these sentences, we prompt the GPT-3 model with instructions tailored for each inference rule. The prompt schema, as depicted in Figure 3, comprise three crucial components:

*Definition* provides a detailed explanation of the task and offers a natural language representation of the reasoning pattern for which we are generating sentences.

*Examples* provide sample sentences that need to be generated. We also illustrate how these sentences will be utilized in later stages, emphasizing the importance of coherence and the inclusion of relevant ontological concepts.

*Format* We provide specific formatting instructions to guide the generation of sentences.

An example of a prompt corresponding to the 'Modus Tollens' from PL is presented in Appendix A for better illustration. Note that our objective at this stage is not to generate logical sentences but rather to generate a diverse and coherent set of sentences that encompass various concepts. We also create a negation sentence corresponding to each generated sentence[2]. In this work, the scope of generating negations is simple (refer to Appendix C for examples), however, negations can be more complicated in the case of logic. These generated sentences will be combined with logical connectives in a later stage to form context and questions.

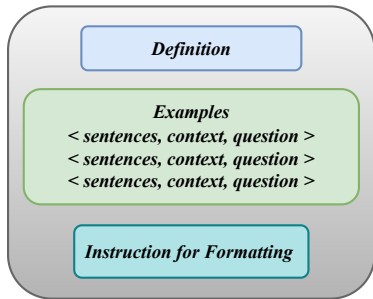

Figure 3: Schematic representation of prompt.

### 2.2.2 NL Conversion

We focus on leveraging the formal expressions of reasoning patterns to create templates that establish the desired NL formulation for each logical connective. For instance, implication: "$p \rightarrow q$" is expressed as "If $p$, then $q$", conjunction: "$p \wedge q$" is expressed as "$p$ and $q$.", and disjunction: "$p \vee q$" is expressed as "At least one of the following is true: (1) $p$ and (2) $q$. Note that we do not know which of (1) and (2) is true. It is possible that only (1) is true, or only (2) is true, or both are true."

With these established formulations, we proceed to utilize the sentences generated in §2.2.1 to create the context and questions corresponding to reasoning patterns. For instance, let's consider the "Modus Tollens" from PL $(((p \rightarrow q) \wedge \neg q) \vdash \neg p)$, and the "Bidirectional Dilemma" from FOL $(\forall x((p(x) \rightarrow q(x)) \wedge (r(x) \rightarrow s(x))) \wedge (p(a) \vee \neg s(a))) \vdash (q(a) \vee \neg r(a)))$. Table 3 presents examples of logical context and questions for these inference rules, and Appendix C showcases further examples corresponding to each inference rule and patterns from *LogicBench*.

### 2.2.3 Variation Generation

After generating the context and questions in §2.2.2, we generate semantically preserving and inverting variations of questions. Let's consider the example of "Modus Tollens" from Table 3, where the question is: "If he won't order pizza for dinner, does this imply that Liam didn't finish his work early?" In this question, we observe two propositions: $s_1$, representing the statement "Liam didn't finish his work early," and $s_2$, representing the statement "He won't order pizza for dinner." By perturbing these propositions, we can create four possible tuples: $< s_1, s_2 >, < \neg s_1, s_2 >, < s_1, \neg s_2 >, < \neg s_1, \neg s_2 >$. Each tuple represents a combination of true or negation values for the propositions. Although it is possible to create more combinations from $< s_1, \neg s_1 >$, and $< s_2, \neg s_2 >$, we refine and restrict the set of triplets to exclude those that undermine the validity of the inference rule. To generate question variations, we replace the propositions in the original question with the corresponding tuples from the generated variations, hence, adding more diversity to *LogicBench*. This process allows us to create different variations of the question, as illustrated in Figure 2 (Step 3). More examples of question variations are in Appendix B.

---

[2]We use `https://github.com/dmlls/negate` to generate negated sentences

| Axiom | Generated sentences in stage 1 | Context and Question |
|---|---|---|
| Modus Tollens | p: Liam finished his work early. 
 ¬p: Liam did not finish his work early. 
 q: He will order pizza for dinner. 
 ¬q: He will not order pizza for dinner. | **Context:** If Liam finished his work early, then he will order pizza for dinner. 

 **Question:** If he won't order pizza for dinner, does this imply that Liam didn't finish his work early? |
| Bidirectional Dilemma | p(x): someone drinks lots of water 
 q(x): they will feel hydrated 
 r(x): they eat too much sugar 
 s(x): they will experience a sugar crash 
 p(a): Jane drinks lots of water 
 ¬p(a): Jane does not drink lots of water 
 q(a): she will feel hydrated 
 ¬q(a): she will not feel hydrated 
 r(a): she eats too much sugar 
 ¬r(a): she does not eat too much sugar 
 s(a): she will experience a sugar crash 
 ¬s(a): she will not experience a sugar crash | **Context:** If someone drinks lots of water, then they will feel hydrated. If they eat too much sugar, then they will experience a sugar crash. We know that at least one of the following is true (1) Jane drinks lots of water and (2) she won't experience a sugar crash. Note that we do not know which ones of (1) and (2) are true. It might be the case that only (1) is true, or only (2) is true or both are true. 

 **Question:** If at least one of (1) and (2) is true, can we say, at least one of the following must always be true? (a) she will feel hydrated and (b) she doesn't eat too much sugar. |

Table 3: Illustrative examples of logical context and questions created using sentences that are generated in the first stage §2.2.1.

## 2.3 Statistics and Qualitative Analysis

**Statistics** We introduce two versions of our proposed dataset: *LogicBench(Eval)* and *LogicBench(Aug)*. Statistics of both versions are presented in Table 4. Here, *LogicBench(Eval)* is created using the above method along with human-in-loop to ensure the quality of generated data, whereas *LogicBench(Aug)* is only a synthetically augmented version for training purposes.

These two versions aim to accommodate different evaluation and training needs to explore logical reasoning. Considering the cost and complexity associated with recent LLMs such as GPT-3, and GPT-4, we believe that *LogicBench(Eval)* provides a more feasible evaluation benchmark.

| Dataset | # of Instances per Axiom | Total # of Instances | Total # of Instances (Including Variations) |
|---|---|---|---|
| *LogicBench(Eval)* | 20 | 500 | 1720 |
| *LogicBench(Aug)* | 150 | 3750 | 12908 |

Table 4: Statistics of the *LogicBench(Eval)* and *LogicBench(Aug)*

**Quality of Data** Throughout the data generation phase of *LogicBench(Eval)*, the authors conduct a review of the logical formations to ensure they adhered to the intended structure. We examine each reasoning pattern for any potential discrepancies, ensuring that they were logically sound and correctly represented the intended relationships between propositions. In addition to the logical formation, we also dedicated considerable effort to eliminating typos and validating the grammar.

## 3 Results and Analysis

### 3.1 Experimental Setup

**Task Formulation** We formulate binary classification task using *LogicBench* to evaluate the logical reasoning ability of LLMs. Let us consider a set of data instances $\mathcal{I}_{a,L}$ corresponding to axiom $a$ and logic type $L$. In this set, $i^{th}$ instance is represented as $\mathcal{I}^i_{a,L} = \{(c_i, Q_i)\}$ where $c_i$ represents context and $Q_i = \{q_1, q_2, ..., q_n\}$ represents set of question and its variations corresponding to $i^{th}$ instance. As discussed in §2, each context ($c$) represents logical rules (e.g., All cats have fur. Tom is a cat.) and question ($q$) represents the conclusion (e.g., Does Tom have fur?). To each context and question pair, i.e., $< c, q >$, we assign a label from the set $\mathcal{Y} = \{Yes, No\}$. We assign a label $Yes$ if the conclusion logically entails the context, otherwise, assign a label $No$. To evaluate any model on this setup, we provide $< c, q >$ as input to predict a label from $\mathcal{Y}$.

**Experiments**  We evaluate easily available and widely used prompting models (i.e., GPT-3 (davinci-003) and ChatGPT), and instruction-tuned models (FLAN-T5 and Tk-instruct) on *LogicBench(Eval)*. Since logical reasoning is an important aspect of different QA tasks, we also evaluate UnifiedQA. Each model is evaluated in a zero-shot setting where the prompt is provided to the model without any in-context examples. This approach allows us to determine LLM's inherent ability to do logical reasoning (based on pre-training), as we can not expect that various logical inference rules/patterns will always be made part of prompts. However, we do evaluate these models in a few-shot setting, and present the results in Appendix F. We also present exploratory – only exploratory because of the limited availability of their inference APIs – analysis over Bard and GPT-4 in Appendix G.

In addition, we employed the T5-large model and trained it on the *LogicBench(Aug)* resulting in a model named LogicT5. LogicT5 has achieved $\sim 97\%$ of accuracy on *LogicBench(Eval)* since it is evident that supervised fine-tuning improves results by a large margin. Subsequently, we performed fine-tuning on four other logical reasoning datasets: LogiQA, Reclor, LogicNLI, and FOLIO. Our experiments were carried out in two settings: single-task (fine-tuning and evaluation on one dataset) and multi-task (fine-tuning on all four datasets combined, with separate evaluations for each dataset). A detailed experimental setup is described in Appendix D.

**Metrics**  Here, we evaluate performance in terms of accuracy corresponding to each label, i.e., $A(Yes)$ and $A(No)$. We evaluate each model on three different prompts and report average results across these prompts. All prompts used for experiments are described in Appendix D.

## 3.2   Benchmark Results

Table 5 represents label-wise accuracy ($A(Yes)$ and $A(No)$) corresponding to each LLMs. Here, we focus on analyzing the $A(Yes)$ since the aim is to understand the model's logical reasoning capabilities in answering the question where the conclusion entails the logical context. Table 5 provides valuable insights into the performance of different models on various logic types. For PL, UnifiedQA exhibits an average performance of 15%, while FLAN-T5 and Tk-instruct achieve $\sim 25\%$. GPT-3 demonstrates a performance of 57.6%, and ChatGPT achieves 46.8%. Moving on to FOL, these models showcase performance accuracy of 52.7%, 51.2%, 55.7%, 76.2%, and 72.6% for UnifiedQA, FLAN-T5, Tk-instruct, GPT-3, and ChatGPT, respectively. On the NM reasoning, these models show an accuracy of 63.5%, 56.2%, 56.3%, 62%, and 70.9%, respectively. Overall, these models display an average performance of $\sim 34\%$, $\sim 61\%$, and $\sim 62\%$ on PL, FOL, and NL.

From Table 5, we can observe that models struggle more with inference rules of PL compared to FOL and NM reasoning. Furthermore, it is noticeable that each model performs relatively better on questions with a negative response (i.e., $No$) compared to questions with a positive response (i.e., $Yes$). This observation suggests that the models struggle to fully comprehend the logical relationship between the context and the conclusion (i.e., lower $A(Yes)$). However, they demonstrate a relatively stronger understanding when the relationship is contradictory in nature (i.e., higher $A(No)$). However, analyzing the performance of the models on inference rules is crucial to understand their limitations. Table 5 presents the inference rule-wise performance for each model as well.

## 3.3   Analysis and Discussion

**Large models are better logical reasoners.**   Based on the observed performance from Table 5, it becomes evident that larger model sizes and extensive pre-training data contribute to a better understanding of logical aspects. Consequently, models with larger sizes tend to exhibit higher performance across different types of logic. Nonetheless, the average performance remains at around 52.7%, indicating room for improvement in these models' logical comprehension capabilities.

**Negations are hard to understand when embedded with logical rules.**   Regarding PL and FOL, it is apparent that the models struggle more with the DS, DD, and MT inference rules. A closer look at Table 1 reveals that all of these axioms include examples where the models need to draw conclusions based on negated premises. This indicates that the models encounter difficulties when

| Type | Axiom | FLAN-T5 | | Tk-instruct | | UnifiedQA | | GPT-3 | | ChatGPT | |
|---|---|---|---|---|---|---|---|---|---|---|---|
| | | $A(No)$ | $A(Yes)$ | $A(No)$ | $A(Yes)$ | $A(No)$ | $A(Yes)$ | $A(No)$ | $A(Yes)$ | $A(No)$ | $A(Yes)$ |
| PL | HS | 100 | 48.4 | 97.9 | 57.9 | 81.6 | 95.2 | 97.6 | 78.3 | 100 | 57.2 |
| | DS | 64.1 | 8.3 | 67.9 | 10.9 | 68.8 | 2.1 | 75.5 | 33.3 | 73.8 | 5.5 |
| | CD | 50 | 25 | 75 | 25 | 63.3 | 0 | 97.7 | 75.4 | 99.4 | 81.0 |
| | DD | 75 | 25 | 75 | 25 | 71.1 | 0 | 78 | 43.4 | 100 | 33.1 |
| | BD | 75 | 25 | 75 | 25 | 88.8 | 0 | 80.5 | 97 | 97.4 | 58.0 |
| | MT | 92.2 | 44.6 | 74.5 | 24.4 | 74.1 | 22.9 | 72.5 | 17.5 | 92.3 | 45.5 |
| | MI | 63.7 | 23.2 | 64.2 | 0 | 90.3 | 0 | 81.5 | 33.3 | 91.3 | 41.3 |
| | CT | 25 | 16.7 | 78.3 | 31.5 | 95.2 | 0 | 95.8 | 97 | 100 | 52.3 |
| | **Avg** | **68.1** | **27** | **76** | **25** | **79.1** | **15** | **84.9** | **59.4** | **94.3** | **46.8** |
| FOL | EI | 100 | 100 | 95 | 100 | 98.4 | 100 | 88.9 | 100 | 89.7 | 100 |
| | UI | 98.1 | 86.9 | 89.3 | 84.4 | 72.5 | 94.9 | 88.2 | 98.2 | 85.1 | 94.3 |
| | MP | 99.2 | 79.3 | 88.6 | 86.3 | 70.7 | 87.4 | 81.6 | 82.3 | 88.5 | 80.1 |
| | HS | 100 | 49.2 | 100 | 52.7 | 83.6 | 88.3 | 94.9 | 78.7 | 95.7 | 53.1 |
| | DS | 72.1 | 21.9 | 71.4 | 4.6 | 80.4 | 55.6 | 81.8 | 96.3 | 88.2 | 97.6 |
| | CD | 75 | 25 | 91.7 | 62 | 54.6 | 0 | 93.2 | 65.9 | 93.7 | 87.9 |
| | DD | 75 | 25 | 87.4 | 28 | 94.4 | 0 | 75.4 | 44.4 | 83.9 | 30.6 |
| | BD | 25 | 25 | 91.7 | 47 | 100 | 33.3 | 77.5 | 94.4 | 98.7 | 67.6 |
| | MT | 93.3 | 48.1 | 81.8 | 35.9 | 70.8 | 15.2 | 74.8 | 25.7 | 85.9 | 42.3 |
| | **Avg** | **82** | **51.2** | **88.5** | **55.7** | **80.6** | **52.7** | **84.1** | **76.2** | **89.9** | **72.6** |
| NM | DRI | 60.5 | 59.6 | 52.5 | 53.8 | 58.2 | 61.7 | 75 | 100 | 75.6 | 89.6 |
| | DRS | 66.3 | 2.9 | 60 | 3.9 | 67.3 | 2.8 | 72.6 | 10.1 | 72.7 | 0 |
| | DRD | 95 | 95 | 88.8 | 75.7 | 68.1 | 97.8 | 84.7 | 100 | 82.2 | 100 |
| | DRO | 40 | 42.6 | 43.8 | 45.3 | 53.2 | 91.7 | 65.3 | 100 | 70.3 | 100 |
| | RE1 | 74.2 | 24.2 | 85.2 | 28 | 75.8 | 33.3 | 74.3 | 0 | 81.4 | 33.6 |
| | RE2 | 100 | 100 | 98.2 | 93.8 | 56.2 | 66.7 | 50 | 0 | 62.3 | 64.7 |
| | RE3 | 65.6 | 63 | 78.3 | 57.7 | 78.2 | 81 | 64.5 | 93.6 | 67.2 | 82.7 |
| | RAP | 70.1 | 62.6 | 76.9 | 92.5 | 64.5 | 73 | 56.8 | 92.2 | 58.3 | 96.9 |
| | **Avg** | **71.5** | **56.2** | **73** | **56.3** | **65.2** | **63.5** | **67.9** | **62** | **71.3** | **70.9** |

Table 5: Evaluation of LLMs in terms of label-wise accuracy on LogicBench(Eval), where $A(Yes)$ and $A(No)$ denote the accuracy for the $Yes$ and $No$ labels, respectively. DRI: Default Reasoning with Irrelevant Information, DRS: Default Reasoning with Several Defaults, DRD: Default Reasoning with a Disabled Default, DRO: Default Reasoning in an Open Domain, RE1: Reasoning about Unknown Expectations I, RE2: Reasoning about Unknown Expectations II, RE3: Reasoning about Unknown Expectations III, RAP: Reasoning about Priorities

negated premises are introduced. Additionally, the performance of the models tends to decrease when inference rules involve negations.

**Longer inference rules are still challenging.** Table 1 indicates that the models face challenges when handling longer rules, such as BD, CD, and DD, both in PL and FOL. Hence, it can be concluded that these models struggle with longer logical dependencies in the premise, particularly when a higher number of propositions are present. In the case of NM reasoning, the models exhibit lower performance in DRS of NM reasoning, indicating that a higher number of rules in the context often leads to more frequent mistakes.

**Effect on other logic datasets** Table 6 represents the accuracy comparison between LogicT5 and baseline T5-large in both single-task and multi-task settings. The results indicate that training LLMs on *LogicBench(Aug)* has a greater impact on logic datasets that primarily focus on logical reasoning, such as FOLIO and LogicNLI. Hence, we can observe that LogicT5 consistently outperforms the baseline for LogicT5 and FOLIO. However, LogiQA and ReClor encompass other forms of reasoning in addition to logical reasoning, hence, LogicT5 demonstrates competitive performance on them.

**How do LLMs reason step-by-step?** We investigate the fraction of low-performing axioms that contain various types of logical reasoning steps to predict the answer, and whether the correctness of those steps is correlated with the performance. Here, we perform a case study on ChatGPT. We prompt ChatGPT to generate reasoning steps along with predictions. For PL, we observe that

| Methods | Models | LogiQA | FOLIO | LogicNLI | ReClor |
|---|---|---|---|---|---|
| Single-Task | T5-large | 16.8 | 69.6 | 82.3 | 35.4 |
| | LogicT5 | **16.9** | **71.2** | **84.4** | **36.8** |
| Multi-Task | T5-large | **21.8** | 83.8 | 68.2 | **42.8** |
| | LogicT5 | 19.7 | **85.6** | **69.8** | 40.0 |

Table 6: Performance comparison between LogicT5 and baseline T5-large in terms of accuracy.

while the model can effectively reason the initial section of the *disjunctive syllogism* involving two possibilities $p$ or $q$, it encounters challenges in deducing whether $q$ should follow from the $\neg p$. For FOL, ChatGPT encounters challenges in comprehending longer logical contexts, resulting in a lack of confidence in establishing the relationship between given propositions. Furthermore, to derive an accurate conclusion when the rules are followed correctly, the model relies on supplementary evidence. We observe that ChatGPT encounters difficulties in comprehending the nuanced meanings of words such as "usually", "normally" and "typically" when establishing sentence relationships within NM reasoning. Notably, when it comes to the rule of default reasoning, ChatGPT fails to grasp inherent associations between two entities that commonly share characteristics. Examples and more analysis of generated explanations for each logic type are presented in Appendix E.

## 4 Related Work

LogiQA [11] and ReClor [19] have made notable contributions by compiling multichoice questions from standardized graduate admission examinations that demand diverse forms of logical reasoning. However, in contrast to our LogicBench, these datasets involve complex mixed forms of reasoning and do not specifically focus on assessing logical reasoning in isolation. A few past attempts have been made to create datasets to evaluate only logical reasoning while excluding other forms of reasoning. For example, CLUTTER [15] covers inductive reasoning, [5] covers temporal logic, and Ruletaker [4] evaluates whether a transformer-based model emulates deductive reasoning over synthetically generated statements in a limited setting. LogicNLI [16] introduced a diagnostic benchmark for FOL reasoning, with the dataset constructed by first automatically generating logic expressions and then replacing the entity and attribute placeholders in the logic expressions with simple and random subjects and predicates. FOLIO [6] gives diverse and complex logical expressions, however, it is only limited to FOL. ProntoQA [14] provides explanation and reasoning steps but is limited to modus ponens in FOL. Additional datasets for evaluating logical reasoning also exist such as TaxiNLI [7] introduce logical taxonomy in NLI task and RuleBert [13] covers only soft logical rules. In summary, LogicBench is evaluate logical reasoning in isolation and provides more diverse inference rules and logic types compared to existing datasets. Extended related work is discussed in Appendix H.

## 5 Conclusions

To study the logical reasoning ability of LLMs, we introduced a novel benchmark called *LogicBench* which consists of 25 distinct inference rules and reasoning patterns covering propositional, first-order, and non-monotonic logics. We released two versions of the dataset: *LogicBench(Eval)* and *LogicBench(Aug). LogicBench(Eval)* serves as a high-quality, cost-effective, and reliable dataset for evaluating LLMs, while *LogicBench(Aug)* can be utilized for training purposes. Through comprehensive experiments, we showed that models such as GPT-3 and ChatGPT do not perform well on *LogicBench*, even though they require the application of only a single inference rule in positive (i.e., label 'Yes') data instance. Furthermore, we demonstrated that LLMs trained using *LogicBench(Aug)* showcase an improved understanding of logical reasoning, resulting in a better performance on existing logic datasets. Though *LogicBench* facilitates the evaluation and improvement of the logical reasoning ability of LLMs, it can be further extended by incorporating other inference rules and logic types; and having data instances that require applications of multiple inference rules.

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
