# LogicBench: A Benchmark for Evaluation of Logical Reasoning (Supplementary Material)

## A    Example Prompt for Sentence Generation

Figure 1 illustrates an example prompt for the inference rule, namely, 'modus tollens' from propositional logic (PL). Modus tollens is formally represented as $((p \rightarrow q) \land \neg q) \vdash \neg p$, which can be understood in natural language as "If $p$ implies $q$, and we know $\neg q$, then we can conclude $\neg p$." In this prompt, the definition provides a comprehensive description of the inference rule in natural language. To encourage the generation of more relevant and coherent sentences, the prompt includes an examples section that demonstrates how the generated sentences will be utilized in a later stage. This serves, as an illustration, to guide GPT-3 in producing suitable outputs. In Figure 1, we present three examples involving sentences $p$ and $q$, along with their respective contexts and questions. The prompt also includes instructions on how the generated sentences should be formatted.

## B    Examples of Data Instances

This section provides examples of *(context, question, answer)* tuples corresponding to each inference rule and reasoning pattern. Additionally, it highlights the diverse range of question variations within the dataset associated with each inference rule and reasoning pattern.

### B.1    Word Cloud

Figure 2 provides a word cloud derived from the $LogicBench(Eval)$. This word cloud highlights the logical nature and diversity of our evaluation dataset. Words such as 'if', 'normally', 'usually', and 'then' are prominently featured, suggesting their frequent use in the dataset, and

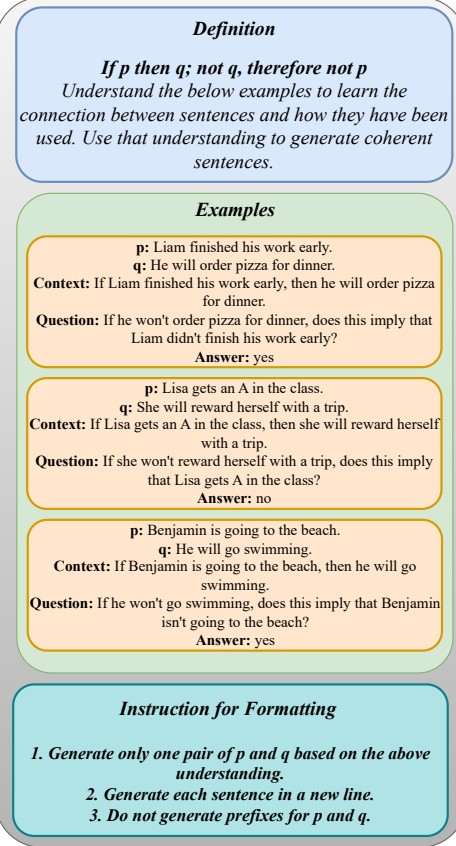

Figure 1: Example prompt for *Modus Tollens* inference rule from PL.

Submitted to the 37th Conference on Neural Information Processing Systems (NeurIPS 2023) Track on Datasets and Benchmarks. Do not distribute.

r

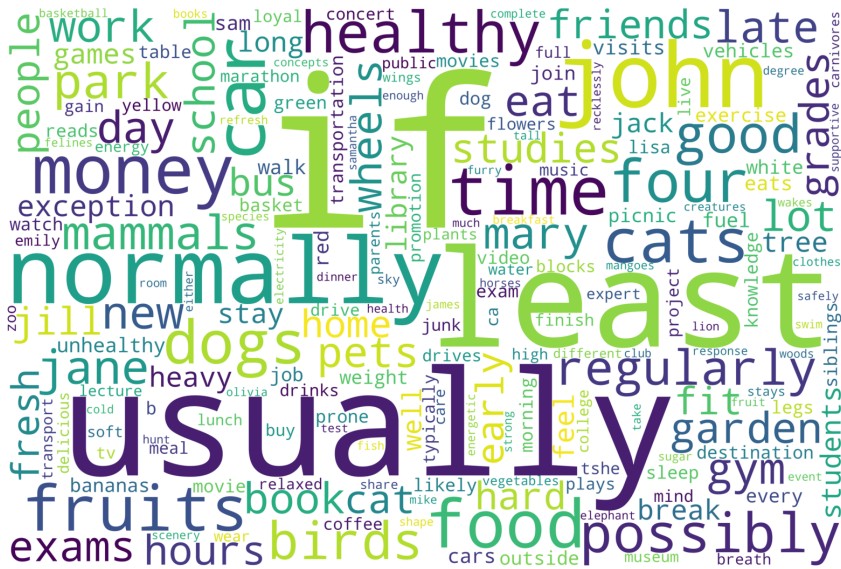

Figure 2: Word cloud of context present in the LB(eval)

suggesting the logical nature of the dataset. Moreover, we can also observe several words consisting
of different ontologies such as 'cat', 'car', 'garden', and many more, suggesting diversity in the
dataset.

## B.2 Propositional Logic (PL)

Here, we discuss examples of each inference rule present in the PL of the *LogicBench* as shown in
Table 1. Table 1 has context related to the inference rule and different variations of the question
according to the rule. For instance, the first row of Table 1 shows the example for inference rule,
*Hypothetical Syllogism (HS)*, formally expressed as $((p \rightarrow q) \land (q \rightarrow r)) \vdash (p \rightarrow r)$. The context
represents the premise, i.e., $((p \rightarrow q) \land (q \rightarrow r))$, and the first question (Q1) represents the
conclusion, i.e., $p \rightarrow r$. Hence, Q1 is labeled as "Yes" since it supports the conclusion given the
logical context. Furthermore, Q2 to Q4 represent different variations of the question by utilizing
the variables $(p, \neg p, r, \neg r)$. For the HS, given the provided context, Q2 to Q4 contain the variations
$\neg p \rightarrow r$, $p \rightarrow \neg r$, and $\neg p \rightarrow \neg r$, respectively, and are labeled as "No" since they do not support the
conclusion.

## B.3 First-Order Logic (FOL)

Here, we discuss examples of each inference rule and two axioms (i.e., Existential Instantiation and
Universal Instantiation) present in the FOL from the *LogicBench* as shown in Table 2. *Existential
Instantiation (EI)*, formally expressed as $\exists x P(x) \Rightarrow P(a)$ indicates that there is an element $a$ in the
domain for which $P(a)$ is true if we know that $\exists x P(x)$ is true. *Universal Instantiation* formally
expressed as $\forall x A \Rightarrow A\{x \mapsto a\}$ indicates that a statement holds true for all instances (x) within a
specific category $A$, hence it is also true for specific instance $a$.

Table 2 represents context related to the inference rule and variations of the question. The process of
generating data instances for FOL follows a similar approach to that of PL. For example, the first
row of Table 2 shows the example for axiom, *Existential Instantiation (EI)*, formally expressed as
$\exists x P(x) \Rightarrow P(a)$. The context represents the initial premise $\exists x P(x)$ and the first question (Q1)
represents the conclusion, i.e., $P(a)$. Hence, Q1 is labeled as "Yes" since it supports the conclusion
given the logical context. Furthermore, we generate the only variant of the question based on $\neg P(a)$
and labeled it as $No$ since it does not support the conclusion.

## B.4 Non-Monotonic (NM) Reasoning

Here, we discuss examples of each reasoning pattern present in the NM reasoning from the *LogicBench* as shown in Table 3. Table 3 has context related to the reasoning pattern and different variants of the question. For example, the first row of Table 3 shows the example for *Default Reasoning with Irrelevant Information (DRI)*. For this reasoning, based on the given context, there are also two possible variations of the question where one with a correct conclusion labeled as $Yes$ and another with an incorrect conclusion labeled as $No$.

| Rules | Context | Question |
|---|---|---|
| HS | If Jim cleaned his room, then he will get a reward. If he will get a reward, then he will buy a new toy. | **Q1:** If Jim cleaned his room, does this imply that he will buy a new toy? (Yes) 
 **Q2:** If Jim didn't clean his room, does this entail that he won't buy a new toy? (No) 
 **Q3:** If Jim cleaned his room, does this imply that he won't buy a new toy? (No) 
 **Q4:** If Jim didn't clean his room, does this imply that he will buy a new toy? (No) |
| DS | We know that at least one of the following is true (1) Chloe is studying for her exams and (2) Mila is going on vacation. Note that we do not know which ones of (1) and (2) are true. It might be the case that only (1) is true, or only (2) is true or both are true. | **Q1:** If Chloe isn't studying for her exams, does this entail that Mila is going on vacation? (Yes) 
 **Q2:** If Chloe isn't studying for her exams, does this mean that Mila isn't going on vacation? (No) 
 **Q3:** If Chloe is studying for her exams, does this imply that Mila isn't going on vacation? (No) 
 **Q4:** If Chloe is studying for her exams, does this imply that Mila is going on vacation? (No) |
| CD | If I go for a walk, then I will get some fresh air. If I stay home, then I will watch a movie. We know that at least one of the following is true (1) I go for a walk and (2) I stay home. Note that we do not know which ones of (1) and (2) are true. It might be the case that only (1) is true, or only (2) is true or both are true. | If at least one of (1) and (2) is true, can we say, at least one of the following must always be true? 
 **Q1:** (a) I will get some fresh air and (b) I will watch a movie (Yes) 
 **Q2:** (a) I won't get some fresh air and (b) I will watch a movie (No) 
 **Q3:** (a) I will get some fresh air and (b) I won't watch a movie (No) 
 **Q4:** (a) I won't get some fresh air and (b) I won't watch a movie (No) |
| DD | If I order takeout, then I will save time. If I cook a meal, then I will save money. We know that at least one of the following is true (1) I won't save time and (2) I won't save money. Note that we do not know which ones of (1) and (2) are true. It might be the case that only (1) is true, or only (2) is true or both are true. | If at least one of (1) and (2) is true, can we say, at least one of the following must always be true? 
 **Q1:** (a) I don't order takeout and (b) I don't cook a meal (Yes) 
 **Q2:** (a) I order takeout and (b) I cook a meal (No) 
 **Q3:** (a) I don't order takeout and (b) I cook a meal (No) 
 **Q4:** (a) I order takeout and (b) I don't cook a meal (No) |
| BD | If it rains, then we will stay inside. If it is sunny, then we will go for a walk. We know that at least one of the following is true (1) it rains and (2) we will not go for a walk. Note that we do not know which ones of (1) and (2) are true. It might be the case that only (1) is true, or only (2) is true or both are true. | If at least one of (1) and (2) is true, can we say, at least one of the following must always be true? 
 **Q1:** (a) We will stay inside and (b) it is not sunny (Yes) 
 **Q2:** (a) We will not stay inside and (b) it is sunny (No) 
 **Q3:** (a) We will stay inside and (b) it is sunny (No) 
 **Q4:** (a) We will not stay inside and (b)it is not sunny (No) |
| MT | If Mason left his job, then he will not receive any salary. | **Q1:** If he will receive any salary, does this mean that Mason didn't leave his job? (Yes) 
 **Q2:** If he will receive any salary, does this mean that Mason left his job? (No) 
 **Q3:** If he will not receive any salary, does this imply that Mason didn't leave his job? (No) 
 **Q4:** If he will not receive any salary, does this mean that Mason left his job? (No) |
| MI | If Rohan forgot his lunch, then he will not eat at school. | Based on context, can we say, at least one of the following must always be true? 
 **Q1:** (a) Rohan didn't forget his lunch and (b) he will not eat at school (Yes) 
 **Q2:** (a) Rohan forgot his lunch and (b) he will eat at school (No) 
 **Q3:** (a) Rohan forgot his lunch and (b) he will not eat at school (No) 
 **Q4:** (a) Rohan didn't forget his lunch and (b) he will eat at school (No) |
| CT | We know that at least one of the following is true (1) Tom is an avid reader and (2) he devours books of all genres. Note that we do not know which ones of (1) and (2) are true. It might be the case that only (1) is true, or only (2) is true or both are true. | If at least one of (1) and (2) is true, can we say, at least one of the following must always be true? 
 **Q1:** (a) He devours books of all genres and (b) Tom is an avid reader (Yes) 
 **Q2:** (a) He doesn't devour books of all genres and (b) tom is an avid reader (No) 
 **Q3:** (a) He devours books of all genres and (b) tom isn't an avid reader (No) 
 **Q4:** (a) He doesn't devour books of all genres and (b) tom isn't an avid reader (No) |

Table 1: Examples of context and question-answer pairs for each inference rule of PL. (Yes) and (No) indicates the answer to the given question. HS: Hypothetical Syllogism, DS: Disjunctive Syllogism, CD: Constructive Dilemma, DD: Destructive Dilemma, BD: Bidirectional Dilemma, MT: Modus Tollens, MI: Material Implication, CT: Commutation

## C  Examples of NL Conversion

This section illustrates the way natural language logical context and questions are created using the generated sentence in Stage 1. Table 4 shows examples of how context and question are generated from sentences corresponding to each inference rule for PL and FOL. Similarly, Table 5 shows examples of NM reasoning. From Table 4, we can see an example of sentence pairs $(p, q)$ and their

| Rules | Context | Question |
|---|---|---|
| UI | All students need to take an exam to complete their degree. Reema is a student. | **Q1:** Does Reema need to take an exam to complete her degree? (Yes)
**Q2:** Does Reema need not to take an exam to complete her degree? (No) |
| EI | James won the marathon race. | **Q1:** Does this imply that someone won the marathon race? (Yes)
**Q2:** Does this mean that no one won the marathon race? (No) |
| MP | If someone is exhausted, then they will take a rest. | **Q1:** If Jack is exhausted, does this entail that he will take a rest? (Yes)
**Q2:** If Jack isn't exhausted, does this imply that he won't take a rest? (No)
**Q3:** If Jack is exhausted, does this entail that he won't take a rest? (No)
**Q4:** If Jack isn't exhausted, does this entail that he will take a rest? (No) |
| HS | If someone buys all the necessary supplies, then they can start the project. If they can start the project, then they will finish it on time. | **Q1:** If Lily bought all the necessary supplies, does this mean that she will finish it on time? (Yes)
**Q2:** If Lily didn't buy all the necessary supplies, does this imply that she won't finish it on time? (No)
**Q3:** If Lily bought all the necessary supplies, does this entail that she won't finish it on time? (No)
**Q4:** If Lily didn't buy all the necessary supplies, does this imply that she will finish it on time? (No) |
| DS | We know that at least one of the following is true (1) they can go to a museum and (2) they can visit a park. Note that we do not know which ones of (1) and (2) are true. It might be the case that only (1) is true, or only (2) is true or both are true. | **Q1:** If Jill can't go to a museum, does this imply that she can visit a park? (Yes)
**Q2:** If Jill can't go to a museum, does this entail that she can't visit a park? (No)
**Q3:** If Jill can go to a museum, does this entail that she can't visit a park? (No)
**Q4:** If Jill can go to a museum, does this imply that she can visit a park? (No) |
| CD | If someone is painting a picture, then they will frame it. If they are writing a story, then they will publish it. We know that at least one of the following is true (1) John is painting a picture. and (2) She is writing a story. Note that we do not know which ones of (1) and (2) are true. It might be the case that only (1) is true, or only (2) is true or both are true. | If at least one of (1) and (2) is true, can we say, at least one of the following must always be true?
**Q1:** (a) He will frame it and (b) She will publish it (Yes)
**Q2:** (a) He won't frame it and (b) She will publish it (No)
**Q3:** (a) He will frame it and (b) She won't publish it (No)
**Q4:** (a) He won't frame it and (b) She won't publish it (No) |
| DD | If someone takes care of her health, then they will be fit and healthy. If they indulge in unhealthy habits, then they will be prone to diseases. We know that at least one of the following is true (1) Jenny won't be fit and healthy and (2) she won't be prone to diseases. Note that we do not know which ones of (1) and (2) are true. It might be the case that only (1) is true, or only (2) is true or both are true. | If at least one of (1) and (2) is true, can we say, at least one of the following must always be true?
**Q1:** (a) Jenny doesn't take care of her health and (b) she doesn't indulge in unhealthy habits (Yes)
**Q2:** (a) Jenny takes care of her health and (b) she indulges in unhealthy habits (No)
**Q3:** (a) Jenny doesn't take care of her health and (b) she indulges in unhealthy habits (No)
**Q4:** (a) Jenny takes care of her health and (b) she doesn't indulge in unhealthy habits (No) |
| BD | If someone drinks lots of water, then they will feel hydrated. If they eat too much sugar, then they will experience a sugar crash. We know that at least one of the following is true (1) Jane drinks lots of water and (2) she won't experience a sugar crash. Note that we do not know which ones of (1) and (2) are true. It might be the case that only (1) is true, or only (2) is true or both are true. | If at least one of (1) and (2) is true, can we say, at least one of the following must always be true?
**Q1:** (a) She will feel hydrated and (b) she doesn't eat too much sugar (Yes)
**Q2:** (a) She won't feel hydrated and (b) she eats too much sugar (No)
**Q3:** (a) She will feel hydrated and (b) she eats too much sugar (No)
**Q4:** (a) She won't feel hydrated and (b) she doesn't eat too much sugar (No) |
| MT | If someone visits the park, then they have to wear a mask. | **Q1:** If he doesn't have to wear a mask, does this imply that John doesn't visit the park? (Yes)
**Q2:** If he doesn't have to wear a mask, does this entail that John visits the park? (No)
**Q3:** If he has to wear a mask, does this imply that John doesn't visit the park? (No)
**Q4:** If he has to wear a mask, does this imply that John visits the park? (No) |

Table 2: Examples of context and question-answer pairs for each inference rule and axiom of FOL. (Yes) and (No) indicates the answer to the given question. EI: Existential Instantiation, UI: Universal Instantiation, MP: Modus Ponens

| Patterns | Context | Question |
|---|---|---|
| DRI | Cats and dogs are mammals. mammals typically have fur. Cats don't have fur. Dogs are loyal animals. | **Q1:** Does this imply that dogs have fur? (Yes)
**Q2:** Does this entail that dogs don't have fur? (No) |
| DRS | John and Mary are parents. Parents are usually loving and supportive. Parents are normally responsible. Mary isn't loving and supportive. John is not responsible. | **Q1:** Does this imply that Mary is responsible and John is loving and supportive? (Yes)
**Q2:** Does this entail that Mary isn't responsible and John is loving and supportive? (No)
**Q3:** Does this imply that Mary is responsible and John isn't loving and supportive? (No)
**Q4:** Does this entail that Mary isn't responsible and John isn't loving and supportive? (No) |
| DRD | Jenny and Anna are tall. Tall people usually play basketball. Anna is possibly an exception to this rule. | **Q1:** Does this entail that Jenny plays basketball? (Yes)
**Q2:** Does this mean that Jenny doesn't play basketball? (No) |
| DRO | Hummingbirds are birds. Birds migrate south for the winter. Hummingbirds do not migrate south for the winter. | **Q1:** Does this mean that all other birds than hummingbirds migrate south for the winter? (Yes)
**Q2:** Does this mean that all other birds than hummingbirds don't migrate south for the winter? (No) |
| RE1 | Cats, dogs, and horses are animals. Animals are usually considered to be intelligent creatures. At least one of the cats or dogs is not considered intelligent. | **Q1:** Does this entail that horses are considered to be intelligent creatures and exactly one of the cats or dogs is not considered intelligent? (Yes)
**Q2:** Does this mean that horses aren't considered to be intelligent creatures and exactly one of cats or dogs is not considered intelligent? (No)
**Q3:** Does this mean that horses are considered to be intelligent creatures and exactly one of cats or dogs is considered intelligent? (No)
**Q4:** Does this implies that horses aren't considered to be intelligent creatures and exactly one of cats or dogs is considered intelligent? (No) |
| RE2 | Cats normally meow. At least one species of cat doesn't meow. | **Q1:** Does this entail that exactly one species of cat doesn't meow? (Yes)
**Q2:** Does this imply that exactly one species of cat meows? (No) |
| RE3 | Cars have four wheels. Wheels normally have spokes. At least one wheel does not have spokes. | **Q1:** Does this imply that cars have four wheels with spokes? (Yes)
**Q2:** Does this mean that cars don't have four wheels with spokes? (No) |
| RAP | John asserts that Sally was in the store. Jane asserts that Sally was not in the store. | **Q1:** If John's evidence is more reliable than Jane's, does this mean that Sally was in the store? (Yes)
**Q2:** If John's evidence is more reliable than Jane's, does this mean that Sally wasn't in the store? (No)
**Q3:** If John's evidence is less reliable than Jane's, does this entail that Sally was in the store? (No)
**Q4:** If John's evidence is less reliable than Jane's, does this imply that Sally wasn't in the store? (Yes) |

Table 3: Examples of context and question-answer pairs for each reasoning pattern of NM Reasoning. (Yes) and (No) indicates the answer to the given question. DRI: Default Reasoning with Irrelevant Information, DRS: Default Reasoning with Several Defaults, DRD: Default Reasoning with a Disabled Default, DRO: Default Reasoning in an Open Domain, RE1: Reasoning about Unknown Expectations I, RE2: Reasoning about Unknown Expectations II, RE3: Reasoning about Unknown Expectations III, RAP: Reasoning about Priorities

corresponding negation pairs $(\neg p, \neg q)$ for the 'modus tollens' inference rule from PL. These pairs are utilized to generate logical context and questions. Similarly, in the second row, we have four generic rules with variable $x$ $(p(x), q(x), r(x), s(x))$ and their specific cases (i.e., $x = a$), along with their respective negative sentence pairs $[(p(a), \neg p(a)), (q(a), \neg q(a)), (r(a), \neg r(a)), (s(a), \neg s(a))]$. These examples demonstrate the generation of logical context and questions for the FOL inference rule called 'Bidirectional Dilemma (BD)', as shown in Table 4. From Table 5, the first row presents an example of context and questions generated from a sentence pair for the 'Default Reasoning with Irrelevant Information (DRI)' from NM reasoning. In this specific instance, the generated sentences are $(p, q, r, s, t)$, and the negation is only required for the sentence $t$. Therefore, there is a single negation pair $(t, \neg t)$, which is used to generate questions specific to the 'DRI'.

# D    Experimental Setup

## D.1    Extended Discussion on Experiments

**Zero-shot setting**    We evaluate GPT-3 (text-davinci-003) and ChatGPT by utilizing their APIs provided by OpenAI[1]. The evaluation is conducted on the versions of GPT-3 and ChatGPT released in April 2023. It's important to note that these models are regularly updated, so when reproducing the results presented in Table 5 (main paper), there is a possibility of variations. For FLAN-T5, Tk-

---

[1] `https://platform.openai.com/docs/guides/gpt`

instruct, and UnifiedQA, we utilize the large, 3b, and 3b versions, respectively, from the huggingface model repository[2].

**Experiments on other logic datasets** In single and multi-task experiments on other logic datasets, we fine-tune the T5-large model for 10 epochs with a batch size of 16, 1024 maximum input length, an adaptive learning rate of $5e - 05$, and an AdamW optimizer for each experiment. All experiments are performed using NVIDIA RTX A6000 GPUs.

## D.2  Prompts

All the experiments conducted in the zero-shot setting were performed using three distinct prompts. The reported results in Table 5 (main paper) represent the average performance across these prompts. The following are the three different prompts utilized in the experiments:

**Prompt 1**  Given the context and question, respond in 'yes' or 'no'.

**Prompt 2**  Answer the given question ONLY in 'yes' or 'no' using logical reasoning ability. DO NOT generate anything as an answer apart from 'yes' and 'no'.

**Prompt 3**  Given context contains rules of logical reasoning in natural language. Answer the given question based on context ONLY in 'yes' or 'no' using logical reasoning ability. DO NOT generate anything as an answer apart from 'yes' and 'no'.

## E  Case Study on Logical Explanations

This section discusses the performance of the different LLMs in a Chain-of-Thought (CoT) setting on the *LogicBench(Eval)*. Here, we provide a prompt for generating logical explanations along with the final prediction. The following is the prompt provided to perform CoT:

> *Given the context that contains rules of logical reasoning in natural language and question, perform step-by-step reasoning to answer the question. Based on context and reasoning steps answer the question ONLY in 'yes' or 'no'. Please use the below format:*
> *Context: [text with logical rules]*
> *Question: [question that is based on context]*
> *Reasoning steps: [generate step-by-step reasoning]*
> *Answer: Yes/No*

We analyze an average performance across $A(Yes)$ and $A(No)$. Table 6 shows the performance for each inference rule and reasoning pattern achieved by GPT-3, and ChatGPT. From Table 6, we can observe that GPT-3 shows a performance drop of $\sim 8\%$, $\sim 5\%$ and $\sim 6\%$ on PL, FOL, and NM reasoning, respectively compared to zero-shot (i.e., Table 5 (main paper)). Furthermore, GPT-3 shows a performance improvement of $\sim 8\%$, for PL and a drop of $\sim 2\%$ and $\sim 17\%$ on FOL, and NM reasoning, respectively.

This section represents a case study carried out on an inference rule of each type of logic where ChatGPT is not able to perform well. Table 7, 8, and 9 represents a case study for *Disjunctive Syllogism (DS)* from PL, *Destructive Dilemma (DD)* from FOL, and *Default Reasoning with Several Default (DRS)* from NM, respectively. From Table 5 (main paper), we can observe that ChatGPT shows poor performance on these inference rules and reasoning patterns, hence, we believe that analysis of logical explanations corresponding to these can give us more insights into the performance of ChatGPT. Here, we prompt the ChatGPT model to generate reasoning steps along with a predicted answer. Table 7, 8, and 9 represents five examples of *(Context, Question, Correct answer, Logical reasoning steps)* pairs generated by ChatGPT.

---
[2]https://huggingface.co/models

## F  Few-shot Experiments

This section discusses the performance of the different LLMs in a few-shot setting on the *Log-icBench(Eval)*. Here, we provide a prompt along with four distinct examples (two examples with $Yes$ and two examples with $No$). Hence, we analyze an average performance across $A(Yes)$ and $A(No)$. Table 11 shows the performance for each inference rule and reasoning patterns achieved by FLAN-T5, Tk-instruct, UnifiedQA, GPT-3, and ChatGPT. For a better comparison between Table 5 (main paper) and Table 11, we provide the average accuracy and standard deviation of Table 5 (main paper) corresponding to each inference rule in Table 10.

From Table 11, we can observe that UnifiedQA shows a performance drop of $\sim 9\%$, $\sim 2\%$ and $\sim 21\%$ on PL, FOL, and NM reasoning, respectively. As known, UnifiedQA [10] is not trained to understand the instructions, hence, adding in-context examples hampers the performance of this model. In contrast, FLAN-T5, GPT-3, and Tk-instruct show improved performance in the few-shot setting compared to zero-shot. As suggested in Lu et al., 2022 [16], prompt and instruction-tuned models are sensitive to in-context examples. Hence, we see performance variations in Table 11 across all models. Specifically, FLAN-T5 improves an average performance by $\sim 12\%$ for both PL and FOL, however, it shows competitive performance on NM reasoning. On the other hand, GPT-3 consistently outperforms zero-shot baselines by $\sim 4\%$, $\sim 6\%$, and $\sim 14\%$ for PL, FOL, and NM reasoning, respectively. Furthermore, Tk-instruct improves an average performance by $\sim 8\%$ for PL, however, it shows competitive performance on FOL and NM reasoning. Interestingly, in-context examples in a few-shot setting hamper the performance of ChatGPT by $\sim 9\%$, and $\sim 2\%$ for PL, and FOL, respectively, compared to zero-shot. However, ChatGPT improves performance by $\sim 4\%$ on NM reasoning. Improved performance in NM reasoning demonstrates that the inclusion of in-context examples enhances the ability of these models to comprehend the nuanced meanings of logical terms such as "usually" and "typically".

## G  Analysis on Bard and GPT-4

This section discusses a case study carried out on a Bard (Google) and GPT-4 (OpenAI) with the subset of the *LogicBench* dataset for each inference rule and reasoning pattern from PL, FOL, and NM reasoning. To evaluate both models, we use below prompt:

"*Given context contains rules of logical reasoning in natural language. Answer the given question based on context ONLY in 'yes' or 'no' using logical reasoning ability. DO NOT generate anything as an answer apart from 'yes' and 'no'.*"

| Logic | Bard | GPT-4 |
|---|---|---|
| PL | 65.0% | 52.9% |
| FOL | 71.1% | 74.6% |
| NM | 15.0% | 60.9% |
| Average | 51.2% | 62.8% |

Table 12: Performance of BARD and GPT-4 on a subset of *LogicBench(Eval)*

**Bard**  Due to the unavailability of Bard developer API during the evaluation timeline of this paper (April 2023), we manually evaluate a carefully selected subset of *LogicBench(Eval)*. We randomly selected five data samples containing *(context, question, 'Yes')* triplets since the goal of this evaluation is to see if the model can identify the relationship between logical rules (context) and conclusion (question). The experiment was conducted on a total of 125 samples by combining samples from all 25 inference rules and axioms. Results are presented in Table 12. Bard performs well on the FOL with the highest $A(Yes)$ of 71.1% while it achieves $A(Yes)$ of 65%, and 15% on PL and NM reasoning, respectively. Bard performs poorly on NM reasoning showing that it struggles on understanding the nuance of logical words such as 'normally', 'usually', and 'typically'.

**GPT-4**  Due to the limited availability of GPT-4 API and the cost associated with each request, we evaluate GPT-4 on a subset of LogicBench(Eval). We selected only *(context, question, 'Yes')* triplets since the goal of this evaluation is to see if the model can identify the relationship between

logical rules and conclusion. Results are presented in Table 12. GPT-4 performs well on the FOL with the highest $A(Yes)$ of 74.6% while it achieves $A(Yes)$ of 52.9%, and 60.9% on PL and NM logic, respectively.

## H    Extended Related Work

As LLMs such as GPT-4, and Bard continue to evolve rapidly, it becomes increasingly crucial to evaluate their diverse language capabilities, as well as those of forthcoming LLMs. Recently, many datasets have been created that evaluate different language understanding skills such as pronoun resolution [29, 13], commonsense reasoning [34], numerical reasoning [4, 27, 23], qualitative reasoning [33, 32], temporal reasoning [45], and feasibility reasoning [6]. Now, we present the advancements in prompt and instruction tuning using LLMs.

**Prompt Learning**    The introduction of LLMs has significantly shifted the research trend in NLP to prompt-based learning methodologies [15]. Many studies have been conducted to investigate the efficacy of prompt-based learning in various applications including Text classification [43], Natural Language Inference (NLI) [31], and Question Answering (QA) [9], Information Extraction (IE) [2, 3], to name a few. In a recent development, the T0 model employs prompts to achieve zero-shot generalization across various NLP tasks [30]. Scao et al. 2021 suggested that the use of prompts could be as valuable as hundreds of data points on average [12].

**Instruction Learning**    Efrat et al., 2020 [5] was focused on whether existing LLMs understand instructions. The same work in the field of instruction by [8, 42, 7, 44] has been proposed to show that models follow natural language instructions. In addition, Weller et al., 2020 [39] developed a framework focusing on NLP systems that solve challenging new tasks based on their description. Mishra et al., 2021 [22] have proposed natural language instructions for cross-task generalization of LLMs. Similarly, PromptSource [30] and FLAN [38] were built for leveraging instructions and achieving zero-shot generalization on unseen tasks. Moreover, Parmar et al., 2022 [26] shows the effectiveness of instructions in multi-task settings for the biomedical domain. Furthermore, Mishra et al., 2021 [21] discuss the impact of task instruction reframing. Min et al., 2021 [20] introduce a framework to better understand in-context learning. Ouyang et al., 2022 [25] propose the InstructGPT model that is fine-tuned with human feedback to follow instructions. Wang et al., 2022 [36] has developed an instruction-based multi-task framework for few-shot Named Entity Recognition (NER) tasks. In addition, many approaches have been proposed to improve model performance using instructions [40, 14, 37, 17, 11, 28, 24].

**Logic and NLI Datasets**    FraCas [1] offers a unique approach to temporal semantics by converting syntax trees into logical formulas tailored for inference, emphasizing temporal elements such as references, adverbs, aspectual classes, and progressives. The Monotonicity Entailment Dataset (MED) [41] dives deep into monotonicity reasoning within NLI, probing the synergy between lexical and syntactic structures and spotlighting inherent challenges in both upward and downward monotonic reasoning trajectories. The SICK [18] dataset, with its foundation in 10,000 English sentence pairs, is designed to rigorously evaluate semantic relatedness and entailment, leveraging crowdsourced annotations for precision. HANS, or Heuristic Analysis for NLI Systems [19], stands out by rigorously scrutinizing the dependability of NLI models, putting the spotlight on potential pitfalls tied to syntactic heuristics such as lexical overlap. Lastly, CAD [35] introduces a meticulously crafted dataset from Reddit entries, targeting the detection of online abuse. This dataset boasts six distinct primary categories, context-aware annotations, provided rationales, and a rigorous group-adjudication methodology ensuring high-quality annotations.

| Inference Rules | Sentences from Stage 1 | Context and Question |
|---|---|---|
| MT | p: Liam finished his work early.
¬p: Liam did not finish his work early.
q: He will order pizza for dinner.
¬q: He will not order pizza for dinner. | **Context:** If Liam finished his work early, then he will order pizza for dinner.

**Question:** If he won't order pizza for dinner, does this imply that Liam didn't finish his work early? |
| BD | p(x): Someone drinks lots of water.
q(x): They will feel hydrated.
r(x): They eat too much sugar.
s(x): They will experience a sugar crash.
p(a): Jane drinks lots of water.
¬p(a): Jane does not drink lots of water.
q(a): She will feel hydrated.
¬q(a): She will not feel hydrated.
r(a): She eats too much sugar.
¬r(a): She does not eat too much sugar.
s(a): She will experience a sugar crash.
¬s(a): She will not experience a sugar crash. | **Context:** If someone drinks lots of water, then they will feel hydrated. If they eat too much sugar, then they will experience a sugar crash. We know that at least one of the following is true (1) Jane drinks lots of water and (2) She won't experience a sugar crash. Note that we do not know which ones of (1) and (2) are true. It might be the case that only (1) is true, or only (2) is true or both are true.

**Question:** If at least one of (1) and (2) is true, can we say, at least one of the following must always be true? (a) She will feel hydrated and (b) She doesn't eat too much sugar. |
| MP | p(x): Someone is exhausted.
q(x): They will take a rest.
p(a): Jack is exhausted.
¬p(a): Jack is not exhausted.
q(a): He will take a rest.
¬q(a): He will not take a rest. | **Context:** If someone is exhausted, then they will take a rest.

**Question:** If Jack is exhausted, does this entail that he will take a rest? |
| DS | p: John is watching a movie.
¬p: John is not watching a movie.
q: He is playing a game.
¬q: He is not playing a game. | **Context:** We know that at least one of the following is true (1) John is watching a movie and (2) He is playing a game. Note that we do not know which ones of (1) and (2) are true. It might be the case that only (1) is true, or only (2) is true or both are true.

**Question:** If he is not watching a movie, does this mean that is playing a game? |
| HS | p(x): Someone buys all the necessary supplies.
q(x): They can start the project.
r(x): They will finish it on time.
p(a): Lily bought all the necessary supplies.
¬p(a): Lily did not buy all the necessary supplies.
q(a): She can start the project.
¬q(a): She can not start the project.
s(a): She will finish it on time.
¬s(a): She will not finish it on time. | **Context:** If someone buys all the necessary supplies, then they can start the project. If they can start the project, then they will finish it on time.

**Question:** If Lily didn't buy all the necessary supplies, does this imply that she won't finish it on time? |
| CD | p: Harry goes to the park.
¬p: Harry does not go to the park.
q: He will have a picnic with his family.
¬q: He will not have a picnic with his family.
r: He goes to the beach.
¬r: He does not go to the beach.
s: He will swim in the ocean.
¬s: He will not swim in the ocean. | **Context:** If Harry goes to the park, then he will have a picnic with his family. If he goes to the beach, then he will swim in the ocean. We know that at least one of the following is true (1) Harry goes to the park and (2) He goes to the beach. Note that we do not know which ones of (1) and (2) are true. It might be the case that only (1) is true, or only (2) is true or both are true.

**Question:** If at least one of (1) and (2) is true, can we say, at least one of the following must always be true? (a) He will have a picnic with his family and (b) He will swim in the ocean |
| DD | p: I order takeout.
¬p: I did not order takeout.
q: I will save time.
¬q: I will not save time.
r: I cook a meal.
¬r: I did not cook a meal.
s: I will save money.
¬s: I will not save money. | **Context:** If I order takeout, then I will save time. If I cook a meal, then I will save money. We know that at least one of the following is true (1) I won't save time and (2) I won't save money. Note that we do not know which ones of (1) and (2) are true. It might be the case that only (1) is true, or only (2) is true or both are true.

**Question:** If at least one of (1) and (2) is true, can we say, at least one of the following must always be true? (a) I order takeout and (b) I cook a meal |
| CT | p: Tom is an avid reader.
¬p: Tom is not an avid reader.
q: He devours books of all genres.
¬q: He does not devour books of all genres. | **Context:** We know that at least one of the following is true (1) Tom is an avid reader and (2) he devours books of all genres. Note that we do not know which ones of (1) and (2) are true. It might be the case that only (1) is true, or only (2) is true or both are true.

**Question:** If at least one of (1) and (2) is true, can we say, at least one of the following must always be true? (a) he devours books of all genres and (b) Tom is an avid reader |
| MI | p: He is not eating healthy.
¬p: He is eating healthy.
q: He will not gain weight.
¬q: He will gain weight. | **Context:** If he is not eating healthy, then he will not gain weight.

**Question:** Based on context, can we say, at least one of the following must always be true? (a) he is eating healthy and (b) he will gain weight |
| EI | p(x): Someone has coding skills.
p(a): Sheila is a proficient programmer. | **Context:** Sheila is a proficient programmer.

**Question:** Does this mean that someone has coding skills? |
| UI | p(x): Students need to take an exam to complete their degree
p(a): Reema is a student. | **Context:** All students need to take an exam to complete their degree. Reema is a student.

**Question:** Does Reema need to take an exam to complete her degree |

Table 4: Illustrative examples of NL logical context and questions created using sentences that are generated in section 2.2.1 for inference rules and axioms covered in PL and FOL.

| Reasoning Pattern | Sentences from Stage 1 | Context and Question |
|---|---|---|
| DRI | p: Cats and dogs are mammals.
q: Mammals typically have fur.
r: Cats don't have fur.
s: Dogs are loyal animals.
t: Dogs have fur.
¬t: Dogs don't have fur. | **Context:** Cats and dogs are mammals. mammals typically have fur. cats don't have fur. dogs are loyal animals.

**Question:** Does this imply that dogs have fur? |
| DRS | p: John and Mary are parents.
q: Parents are usually loving and supportive.
r: Parents are normally responsible.
s: Mary isn't loving and supportive.
t: John is not responsible.
u: Mary is responsible.
¬u: Mary isn't responsible.
v: John is loving and supportive.
¬v: John isn't loving and supportive. | **Context:** John and Mary are parents. parents are usually loving and supportive. parents are normally responsible. Mary isn't loving and supportive. John is not responsible.

**Question:** Does this imply that Mary is responsible and John is loving and supportive? |
| DRD | p: Jenny and Anna are tall.
q: Tall people usually play basketball.
r: Anna is possibly an exception to this rule.
s: Jenny plays basketball.
¬s: Jenny doesn't play basketball. | **Context:** Jenny and Anna are tall. Tall people usually play basketball. Anna is possibly an exception to this rule.

**Question:** Does this entail that Jenny plays basketball? |
| DRO | p: Hummingbirds are birds.
q: Birds migrate south for the winter.
r: Hummingbirds do not migrate south for the winter.
s: All other birds than hummingbirds migrate south for the winter.
¬s: All other birds than hummingbirds don't migrate south for the winter. | **Context:** Hummingbirds are birds. Birds migrate south for the winter. Hummingbirds do not migrate south for the winter.

**Question:** Does this mean that all other birds than hummingbirds migrate south for the winter? |
| RE1 | p: Cats, dogs, and horses are animals.
q: Animals are usually considered to be intelligent creatures.
r: At least one of the cats or dogs is not considered intelligent.
s: Horses are considered to be intelligent creatures.
¬s: Horses aren't considered to be intelligent creatures.
t: Exactly one of the cats or dogs is not considered intelligent.
¬t: Exactly one of the cats or dogs is considered intelligent. | **Context:** Does this entail that horses are considered to be intelligent creatures and exactly one of cats or dogs is not considered intelligent?

**Question:** Does this entail that horses are considered to be intelligent creatures and exactly one of the cats or dogs is not considered intelligent? |
| RE2 | p: Cats normally meow.
q: At least one species of cat doesn't meow.
r: Exactly one species of cat doesn't meow.
¬r: Exactly one species of cat meows. | **Context:** Cats normally meow. At least one species of cat doesn't meow.

**Question:** Does this entail that exactly one species of cat doesn't meow? |
| RE3 | p: Cars have four wheels.
q: Wheels normally have spokes.
r: At least one wheel does not have spokes.
s: Cars have four wheels with spokes.
¬s: Cars don't have four wheels with spokes. | **Context:** Cars have four wheels. wheels normally have spokes. at least one wheel does not have spokes.

**Question:** Does this imply that cars have four wheels with spokes? |
| RAP | p: John asserts that Sally was in the store.
q: Jane asserts that Sally was not in the store.
r: John's evidence is more reliable than Jane's.
¬r: John's evidence is less reliable than Jane's.
s: Sally was in the store.
¬s: Sally wasn't in the store. | **Context:** John asserts that Sally was in the store. Jane asserts that Sally was not in the store.

**Question:** If John's evidence is more reliable than Jane's, does this mean that Sally was in the store? |

Table 5: Illustrative examples of NL logical context and questions created using sentences that are generated in section 2.2.1 for reasoning patterns covered in NM reasoning.

| Type | Axiom | GPT-3 | | | ChatGPT | | |
|------|-------|-------|-------|-------|-------|-------|-------|
| | | $A(No)$ | $A(Yes)$ | $Acc$ | $A(No)$ | $A(Yes)$ | $Acc$ |
| PL | DD | 80.4 | 37.5 | 67.5 | 89.3 | 58.3 | 80 |
| | HS | 100 | 43.5 | 67.5 | 100 | 66.7 | 87.5 |
| | MT | 90.2 | 41 | 66.2 | 86.4 | 78.6 | 85 |
| | BD | 86.5 | 46.4 | 72.5 | 85.5 | 61.1 | 80 |
| | MI | 100 | 26.7 | 31.2 | 69.5 | 9.5 | 53.8 |
| | DS | 75.4 | 27.3 | 68.8 | 74.7 | 0 | 73.8 |
| | CD | 94.4 | 40.9 | 65 | 91.7 | 75 | 87.5 |
| | CT | 97.8 | 55.9 | 80 | 100 | 55.6 | 80 |
| **Avg** | | **90.6** | **39.9** | **64.8** | **87.1** | **50.6** | **78.4** |
| FOL | DD | 73.8 | 20 | 63.7 | 80.4 | 34.5 | 63.7 |
| | MP | 93.3 | 58 | 71.2 | 81.5 | 80.8 | 81.2 |
| | HS | 100 | 52.6 | 77.5 | 87.9 | 59.1 | 80 |
| | MT | 82.6 | 35.3 | 62.5 | 83.1 | 88.9 | 83.8 |
| | BD | 79.7 | 43.8 | 72.5 | 87.5 | 75 | 85 |
| | UI | 100 | 80 | 87.5 | 83.3 | 100 | 90 |
| | EI | 86.4 | 94.4 | 90 | 62.5 | 100 | 70 |
| | DS | 80.8 | 85.7 | 81.2 | 75 | 0 | 75 |
| | CD | 95.7 | 51.6 | 75 | 90.5 | 82.4 | 88.8 |
| **Avg** | | **88** | **57.9** | **75.7** | **81.3** | **69** | **79.7** |
| NM | RAP | 68.9 | 74.3 | 71.2 | 50 | 0 | 50 |
| | RE3 | 51.9 | 53.8 | 52.5 | 50 | 50 | 50 |
| | RE2 | 48.7 | 0 | 47.5 | 55.6 | 100 | 60 |
| | DRS | 66.1 | 4.2 | 47.5 | 68.8 | 0 | 55 |
| | DRI | 59.1 | 61.1 | 60 | 44.4 | 0 | 40 |
| | RE1 | 72.3 | 13.3 | 61.3 | 73.3 | 0 | 68.8 |
| | DRO | 63.2 | 61.9 | 62.5 | 50 | 0 | 50 |
| | DRD | 87.5 | 59.4 | 65 | 51.3 | 100 | 52.5 |
| **Avg** | | **64.7** | **41** | **58.4** | **55.4** | **31.2** | **53.3** |

Table 6: Performance of GPT-3 and ChatGPT in Chain-of-Thought (CoT) setting in terms of label-wise accuracy on LogicBench(Eval), where A(Yes) and A(No) denote the accuracy for the Yes and No labels, respectively. $Acc$ denotes combined accuracy.

| Context | Question | Answer | Logical Explanation |
|---|---|---|---|
| We know that atleast one of the following is true (1) she is not making a cake for the party and (2) he is writing a report for his boss. Note that we do not know which ones of (1) and (2) are true. It might be the case that only (1) is true, or only (2) is true or both are true. | If she is making a cake for the party, does this mean that he is writing a report for his boss? | Yes | No.
Logical reasoning steps:
1. We are given two possibilities: either she is not making a cake for the party (denoted as statement 1) or he is writing a report for his boss (denoted as statement 2).
2. We are not given any information that directly connects the two statements or suggests that they are dependent on each other.
3. Therefore, the presence or absence of her making a cake does not provide any conclusive information about whether he is writing a report.
4. Without further evidence or a clear relationship between the statements, we cannot infer that if she is making a cake, he must be writing a report.
5. Hence, the answer to the question "If she is making a cake for the party, does this mean that he is writing a report for his boss?" is "No." |
| We know that atleast one of the following is true (1) John is not at school and (2) Mary is taking a break from her job. Note that we do not know which ones of (1) and (2) are true. It might be the case that only (1) is true, or only (2) is true or both are true. | If John is at school, does this entail that Mary is taking a break from her job? | Yes | No, if John is at school, it does not entail that Mary is taking a break from her job.

Logical Reasoning Steps:
1. We are given two possibilities: either John is not at school or Mary is taking a break from her job.
2. If John is at school, it means that the first possibility (John is not at school) is false.
3. Since the first possibility is false, the second possibility (Mary is taking a break from her job) could still be true or false.
4. Therefore, we cannot conclude that Mary is taking a break from her job just because John is at school.
5. The answer to the question is "No." |
| We know that atleast one of the following is true (1) Levi is not studying for his exams and (2) Maya is writing a book. Note that we do not know which ones of (1) and (2) are true. It might be the case that only (1) is true, or only (2) is true or both are true. | If Levi is studying for his exams, does this mean that Maya is writing a book? | Yes | No, if Levi is studying for his exams, it does not necessarily mean that Maya is writing a book.

Logical Reasoning Steps:
1. Given the context, we know that at least one of the following statements is true: a) Levi is not studying for his exams. b) Maya is writing a book.
2. The question asks whether if Levi is studying for his exams, it implies that Maya is writing a book.
3. Since we don't have any information about the relationship between Levi studying for his exams and Maya writing a book, we cannot make a definitive conclusion.
4. It is possible that Levi is studying for his exams while Maya is not writing a book. In this case, the statement "Levi is studying for his exams" does not imply that Maya is writing a book.
5. It is also possible that both statements (1) and (2) are true, meaning Levi is not studying for his exams and Maya is writing a book. In this case, the statement "Levi is studying for his exams" would be false, but it still wouldn't imply that Maya is writing a book.
6. Therefore, without additional information, we cannot determine a direct logical implication between Levi studying for his exams and Maya writing a book. |
| We know that atleast one of the following is true (1) Chloe is studying for her exams and (2) Mila is going on vacation. Note that we do not know which ones of (1) and (2) are true. It might be the case that only (1) is true, or only (2) is true or both are true. | If Chloe isn't studying for her exams, does this entail that Mila is going on vacation? | Yes | No, if Chloe isn't studying for her exams, it does not necessarily entail that Mila is going on vacation. Here are the logical reasoning steps:
1. We are given that at least one of the following is true: (1) Chloe is studying for her exams and (2) Mila is going on vacation.
2. It could be the case that only (1) is true, meaning Chloe is studying for her exams but Mila is not going on vacation.
3. If Chloe isn't studying for her exams, it means that (1) is not true, but it does not provide any information about whether (2) is true or not.
4. Therefore, we cannot conclude that Mila is going on vacation solely based on the information that Chloe isn't studying for her exams.
5. In summary, the absence of Chloe studying for her exams does not imply anything about Mila going on vacation. |
| We know that atleast one of the following is true (1) Ava is not attending the music festival and (2) she is studying for her exams. Note that we do not know which ones of (1) and (2) are true. It might be the case that only (1) is true, or only (2) is true or both are true. | If Ava is attending the music festival, does this mean that she is studying for her exams? | Yes | No.

Logical reasoning steps:
1. We are given that at least one of the following is true: a. Ava is not attending the music festival. b. Ava is studying for her exams.
2. The question asks if Ava is attending the music festival, does this mean she is studying for her exams?
3. Since we know that either (1) or (2) or both are true, it is possible that Ava is attending the music festival without studying for her exams. It could be the case that Ava is not studying for her exams even if she decides to attend the music festival.
4. Therefore, the answer to the question is "No." |

Table 7: Illustrative examples of logical explanations generated by ChatGPT for *Disjunctive Syllogism* from PL

| Context | Question | Answer | Logical Explanation |
|---|---|---|---|
| If someone takes care of her health, then they will be fit and healthy. If they indulge in unhealthy habits, then they will be prone to diseases. We know that atleast one of the following is true (1) Jenny won't be fit and healthy and (2) she won't be prone to diseases. Note that we do not know which ones of (1) and (2) are true. It might be the case that only (1) is true, or only (2) is true or both are true. | If at least one of (1) and (2) is true, can we say, at least one of the following must always be true? (a) Jenny doesn't take care of her health and (b) she doesn't indulge in unhealthy habits | Yes | No, we cannot say that at least one of the following must always be true: (a) Jenny doesn't take care of her health, and (b) she doesn't indulge in unhealthy habits. Logical Reasoning Steps: 1. If someone takes care of their health, they will be fit and healthy. This implies that if Jenny takes care of her health, she will be fit and healthy. 2. If someone indulges in unhealthy habits, they will be prone to diseases. This implies that if Jenny indulges in unhealthy habits, she will be prone to diseases. 3. We are given that at least one of the following is true: (1) Jenny won't be fit and healthy, or (2) she won't be prone to diseases. This means that it could be the case that only (1) is true, only (2) is true, or both are true. 4. If only (1) is true, it means Jenny won't be fit and healthy, but we don't know if she takes care of her health or indulges in unhealthy habits. 5. If only (2) is true, it means Jenny won't be prone to diseases, but we don't know if she takes care of her health or indulges in unhealthy habits. 6. If both (1) and (2) are true, it means Jenny won't be fit and healthy and won't be prone to diseases, but we still don't know if she takes care of her health or indulges in unhealthy habits. 7. Therefore, we cannot conclude that at least one of the following must always be true: (a) Jenny doesn't take care of her health, and (b) she doesn't indulge in unhealthy habits. |
| If someone takes a walk, then they will get fresh air. If they watch tv, then they will feel relaxed. We know that atleast one of the following is true (1) Jill won't get fresh air. and (2) she won't feel relaxed. Note that we do not know which ones of (1) and (2) are true. It might be the case that only (1) is true, or only (2) is true or both are true. | If at least one of (1) and (2) is true, can we say, at least one of the following must always be true? (a) Jill doesn't take a walk. and (b) she doesn't watch tv. | Yes | No, we cannot say that at least one of the following must always be true: (a) Jill doesn't take a walk, and (b) she doesn't watch TV. Logical Reasoning Steps: 1. Given the context, we know that if someone takes a walk, they will get fresh air. 2. We also know that if someone watches TV, they will feel relaxed. 3. It is stated that at least one of the following is true: (1) Jill won't get fresh air, and (2) she won't feel relaxed. 4. Based on the given information, we cannot determine which one of (1) or (2) is true, or if both are true. 5. If (1) is true, it means that Jill won't get fresh air. This does not necessarily imply that she won't take a walk because there could be other reasons for her not getting fresh air. 6. Similarly, if (2) is true, it means that Jill won't feel relaxed. This does not necessarily imply that she won't watch TV because there could be other reasons for her not feeling relaxed. 7. Therefore, we cannot conclude that at least one of the statements (a) Jill doesn't take a walk or (b) she doesn't watch TV must always be true, as the given information does not provide enough evidence to support that conclusion. |
| If someone turns off the lights, then they will save electricity. If they use a generator, then they will pay a high electricity bill. We know that atleast one of the following is true (1) Linda won't save electricity and (2) she won't pay a high electricity bill. Note that we do not know which ones of (1) and (2) are true. It might be the case that only (1) is true, or only (2) is true or both are true. | If at least one of (1) and (2) is true, can we say, at least one of the following must always be true? (a) Linda doesn't turn off the lights and (b) she doesn't use a generator | Yes | No, we cannot say that at least one of the following must always be true: (a) Linda doesn't turn off the lights and (b) she doesn't use a generator. Logical Reasoning Steps: 1. From the given context, we have the following conditional statements: If someone turns off the lights, then they will save electricity. If someone uses a generator, then they will pay a high electricity bill. 2. We are also given that at least one of the following is true: (1) Linda won't save electricity. (2) Linda won't pay a high electricity bill. 3. Let's consider the possibilities: If only (1) is true, it means Linda won't save electricity. This doesn't provide any information about whether Linda will turn off the lights or use a generator. If only (2) is true, it means Linda won't pay a high electricity bill. Again, this doesn't provide any information about whether Linda will turn off the lights or use a generator. 4. If both (1) and (2) are true, it means Linda won't save electricity and won't pay a high electricity bill. This still doesn't provide any information about whether Linda will turn off the lights or use a generator. 5. Since we don't have any specific information about whether Linda will turn off the lights or use a generator, we cannot conclude that at least one of the following must always be true: (a) Linda doesn't turn off the lights and (b) she doesn't use a generator. The given context doesn't establish a direct relationship between Linda's actions regarding turning off the lights or using a generator and statements (1) and (2). |
| If someone reads a lot, then they will get good grades. If they watch television, then they will have a sound sleep. We know that atleast one of the following is true (1) Jill won't get good grades and (2) she won't have a sound sleep. Note that we do not know which ones of (1) and (2) are true. It might be the case that only (1) is true, or only (2) is true or both are true. | If at least one of (1) and (2) is true, can we say, at least one of the following must always be true? (a) Jill doesn't read a lot and (b) she doesn't watch television | Yes | No, we cannot say that at least one of the following must always be true: (a) Jill doesn't read a lot, and (b) she doesn't watch television. Logical Reasoning: 1. If someone reads a lot, then they will get good grades. This implies that if Jill doesn't get good grades, she may not read a lot. This supports the possibility of (a) being true. 2. If someone watches television, then they will have a sound sleep. This implies that if Jill doesn't have a sound sleep, she may not watch television. This supports the possibility of (b) being true. 3. Given the information that at least one of (1) and (2) is true, we cannot determine which specific statement is true. It is possible that only (1) is true, only (2) is true, or both (1) and (2) are true. Therefore, we cannot conclude that either (a) or (b) must always be true. |
| If someone takes a test, then they will get a good grade. If they do not revise enough, then they will not score well. We know that atleast one of the following is true (1) John won't get a good grade and (2) he will score well. Note that we do not know which ones of (1) and (2) are true. It might be the case that only (1) is true, or only (2) is true or both are true. | If at least one of (1) and (2) is true, can we say, at least one of the following must always be true? (a) John doesn't take a test and (b) he revises enough | Yes | No, we cannot say that at least one of the following must always be true: (a) John doesn't take a test, and (b) he revises enough. Here are the logical reasoning steps: 1. Given the context, we know that if someone takes a test, they will get a good grade. This implies the logical statement: "If taking a test, then a good grade." 2. We also know that if someone does not revise enough, they will not score well. This implies the logical statement: "If not revising enough, then not scoring well." 3. The context states that at least one of the following is true: (1) John won't get a good grade, and (2) he will score well. 4. Let's analyze the possibilities: a. If only (1) is true, it means John won't get a good grade. This could be because he either didn't take the test or didn't revise enough. Therefore, (a) John doesn't take a test could be true, but (b) he revises enough that may or may not be true. b. If only (2) is true, it means John will score well. This could be because he either took the test and revised enough or only revised enough. Therefore, (a) John doesn't take a test may or may not be true, but (b) he revises enough could be true. c. If both (1) and (2) are true, it means John won't get a good grade but will still score well. This could happen if he revised enough but didn't take the test. In this case, (a) John doesn't take a test would be true, but (b) he revises enough would be false. 5. Considering all the possibilities, we cannot conclude that at least one of the following must always be true: (a) John doesn't take a test, and (b) he revises enough. The truth values of (a) and (b) depend on the specific circumstances regarding John's test and revision. |

Table 8: Illustrative examples of logical explanations generated by ChatGPT for *Destructive Dilemma* from FOL

| Context | Question | Answer | Logical Steps |
|---|---|---|---|
| John and Mary are parents. Parents are usually loving and supportive. parents are normally responsible. Mary isn't loving and supportive. John is not responsible. | Does this imply that Mary is responsible and John is loving and supportive? | Yes | No, this context does not imply that Mary is responsible and John is loving and supportive.

Logical Reasoning Steps:
1. The context states that parents are usually loving and supportive, and parents are normally responsible.
2. However, it also mentions that Mary isn't loving and supportive and John is not responsible.
3. Since the context specifies that Mary isn't loving and supportive, it contradicts the assumption that Mary is responsible.
4. Similarly, the context specifies that John is not responsible, which contradicts the assumption that John is loving and supportive.
5. Therefore, based on the given context, we cannot conclude that Mary is responsible or that John is loving and supportive. |
| Apples and bananas are fruits. Fruits are usually sweet. Fruits are normally juicy. Apples are not sweet. Bananas are not juicy. | Does this mean that bananas are sweet and apples are juicy? | Yes | No.

Logical Reasoning Steps:
1. The context establishes that fruits are usually sweet and normally juicy.
2. It also states that apples are not sweet and bananas are not juicy.
3. Therefore, based on the context, we can conclude that neither apples nor bananas fit the typical characteristics of fruits.
4. Since the given context does not provide any information suggesting that bananas are sweet or that apples are juicy, we cannot infer that bananas are sweet or that apples are juicy.
5. Therefore, the answer to the question is "No." |
| Cats and dogs are animals. Animals have four legs. Animals have fur. Cats have no fur. Dogs are not four-legged. | Does this mean that dogs have fur and cats have four legs? | Yes | No, dogs do not necessarily have fur and cats do not necessarily have four legs. The given context states that animals have four legs and fur, but it also specifically mentions that cats have no fur and dogs are not four-legged. Therefore, based on the context, we cannot conclude that dogs have fur or that cats have four legs.

Logical reasoning steps:
1. Cats and dogs are animals.
2. Animals have four legs.
3. Animals have fur.
4. Cats have no fur (from the context).
5. Dogs are not four-legged (from the context).
6. Therefore, based on the given context, we cannot conclude that dogs have fur or that cats have four legs. |
| Birds and fish are animals. Animals have four legs. Animals are normally omnivores. Birds don't have four legs. Fish is not an omnivore. | Does this imply that fish have four legs and birds are omnivores? | Yes | No, this context does not imply that fish have four legs and birds are omnivores.

Logical Reasoning Steps:
1. The context states that animals have four legs. This implies that all animals, including birds and fish, are expected to have four legs.
2. However, the context also mentions that birds don't have four legs. This means that birds deviate from the general expectation of having four legs.
3. The context further states that animals are normally omnivores. This implies that the general expectation is for animals to be omnivores, including birds and fish.
4. However, the context also mentions that fish is not an omnivore. This means that fish deviates from the general expectation of being omnivores.
5. Combining these statements, we can conclude that birds do not have four legs and fish is not an omnivore, contradicting the general expectations mentioned in the context.
6. Therefore, based on the given context, it cannot be inferred that fish have four legs and birds are omnivores. |
| Blocks A and B are heavy. Heavy blocks are normally located on the table. Heavy blocks are normally red. A is not on the table. B is not red. | Does the context imply that B is on the table and A is red in color? | Yes | No, the context does not imply that B is on the table and A is red in color.

Logical reasoning steps:
1. The context states that heavy blocks are normally located on the table, but it does not specifically mention the current location of block B. Therefore, we cannot conclude that B is on the table.
2. The context also mentions that heavy blocks are normally red, but it states that block B is not red. Therefore, we cannot conclude that A is red in color.
3. Based on the given information, we can only infer that A is not on the table and B is not red. Any further conclusions about the specific location of B or the color of A cannot be made with certainty. |

Table 9: Illustrative examples of logical explanations generated by ChatGPT for *Default Reasoning with Several Default* from NM reasoning

| Type | Axiom | FLAN-T5 | Tk-instruct | UnifiedQA | GPT-3 | ChatGPT |
|---|---|---|---|---|---|---|
| PL | HS | $73.3_{0.0072}$ | $81.2_{0.0375}$ | $82.1_{0.0505}$ | $91.7_{0.0361}$ | $81.2_{0.0125}$ |
| | DS | $34.6_{0.1449}$ | $47.9_{0.0753}$ | $54.6_{0.0191}$ | $75.4_{0.01909}$ | $69.6_{0.01909}$ |
| | CD | $31.7_{0.0629}$ | $40.8_{0.1264}$ | $35.8_{0.0887}$ | $90_{0.0433}$ | $93.8_{0.0216}$ |
| | DD | $62.5_{0.0433}$ | $67.5_{0.0866}$ | $6.7_{0.0288}$ | $73.3_{0.0072}$ | $49.2_{0.0688}$ |
| | BD | $28.3_{0.0144}$ | $50.4_{0.1120}$ | $12.9_{0.0144}$ | $81.2_{0.0433}$ | $81.2_{0.0216}$ |
| | MT | $70_{0.0125}$ | $56.2_{0.025}$ | $58.3_{0.0144}$ | $56.7_{0.0361}$ | $70.8_{0.0191}$ |
| | MI | $28.7_{0.0125}$ | $45_{0.0433}$ | $26.2_{0}$ | $60.4_{0.0361}$ | $66.7_{0.0473}$ |
| | CT | $41.7_{0.2886}$ | $47.1_{0.2093}$ | $12.1_{0.0505}$ | $95.8_{0.0191}$ | $77.1_{0.0315}$ |
| | **Avg** | $\mathbf{46.4_{0.0733}}$ | $\mathbf{54.5_{0.0894}}$ | $\mathbf{36.1_{0.0333}}$ | $\mathbf{73.5_{0.0283}}$ | $\mathbf{73.7_{0.0302}}$ |
| FOL | EI | $100_{0}$ | $95_{0.0433}$ | $99.2_{0.0144}$ | $93.3_{0.0577}$ | $94.2_{0.0288}$ |
| | UI | $91.7_{0.0288}$ | $86.7_{0.0144}$ | $79.2_{0.0288}$ | $92.5_{0.025}$ | $89.2_{0.0144}$ |
| | MP | $89.6_{0.0072}$ | $86.2_{0.0216}$ | $73.3_{0.0144}$ | $81.7_{0.0144}$ | $85_{0.0216}$ |
| | HS | $74.2_{0.0072}$ | $76.7_{0.0072}$ | $84.2_{0.0382}$ | $90.4_{0.0260}$ | $77.5_{0.0216}$ |
| | DS | $51.7_{0.1993}$ | $61.3_{0.0331}$ | $76.7_{0.0144}$ | $82.9_{0.0382}$ | $89.6_{0.0072}$ |
| | CD | $33.3_{0.0382}$ | $73.8_{0.3473}$ | $15.8_{0.0402}$ | $83.8_{0.0451}$ | $92.1_{0.0191}$ |
| | DD | $55_{0.05}$ | $40.4_{0.0260}$ | $4.2_{0.0260}$ | $75_{0.0125}$ | $54.2_{0.0591}$ |
| | BD | $25.8_{0.0144}$ | $64.2_{0.3068}$ | $5.8_{0.0260}$ | $77.9_{0.0191}$ | $87.1_{0.0439}$ |
| | MT | $73.3_{0.0144}$ | $62.5_{0.0451}$ | $54.2_{0.0072}$ | $67.9_{0.0402}$ | $70_{0.0866}$ |
| | **Avg** | $\mathbf{66.1_{0.0399}}$ | $\mathbf{71.9_{0.0939}}$ | $\mathbf{54.7_{0.0233}}$ | $\mathbf{82.8_{0.0309}}$ | $\mathbf{82.1_{0.0336}}$ |
| NM | DRI | $60_{0.025}$ | $53.3_{0.0803}$ | $56.7_{0.0382}$ | $83.3_{0.0144}$ | $80.8_{0.0288}$ |
| | DRS | $48.3_{0.0072}$ | $37.1_{0.0520}$ | $50_{0.0451}$ | $64.2_{0.0402}$ | $66.7_{0.0144}$ |
| | DRD | $95_{0}$ | $79.2_{0.0946}$ | $71.7_{0.0722}$ | $90.8_{0.0382}$ | $89.2_{0.0144}$ |
| | DRO | $41.7_{0.0382}$ | $44.2_{0.0764}$ | $54.2_{0.0144}$ | $73.3_{0.0382}$ | $76.7_{0.1665}$ |
| | RE1 | $55.8_{0.0617}$ | $41.7_{0.0260}$ | $75_{0.0125}$ | $72.5_{0.025}$ | $60.4_{0.0732}$ |
| | RE2 | $100_{0}$ | $95.8_{0.0382}$ | $54.2_{0.1377}$ | $50_{0}$ | $63.3_{0.1010}$ |
| | RE3 | $64.2_{0.0289}$ | $61.7_{0.0722}$ | $79.2_{0.0144}$ | $71.7_{0.0144}$ | $72.5_{0.05}$ |
| | RAP | $65.4_{0.0473}$ | $82.9_{0.0144}$ | $67.5_{0.0545}$ | $61.7_{0.0072}$ | $64.2_{0.0260}$ |
| | **Avg** | $\mathbf{66.3_{0.0260}}$ | $\mathbf{62_{0.0568}}$ | $\mathbf{63.5_{0.0486}}$ | $\mathbf{70.9_{0.0222}}$ | $\mathbf{71.7_{0.0593}}$ |

Table 10: Average and standard deviation across three different prompts corresponding to each inference rule for Table 5 (main paper).

| Type | Axiom | FLAN-T5 | | Tk-instruct | | UnifiedQA | | GPT-3 | | ChatGPT | |
|------|-------|---------|---------|-------------|---------|-----------|---------|-------|---------|---------|---------|
| | | $A(No)$ | $A(Yes)$ | $A(No)$ | $A(Yes)$ | $A(No)$ | $A(Yes)$ | $A(No)$ | $A(Yes)$ | $A(No)$ | $A(Yes)$ |
| PL | HS | 100 | 50.0 | 100 | 100 | 100 | 34.5 | 98.1 | 70.4 | 100 | 66.6 |
| | DS | 66.6 | 0 | 72.9 | 10.0 | 80.0 | 25.3 | 77.7 | 30.8 | 66.6 | 0 |
| | CD | 87.7 | 56.5 | 75.9 | 100 | 0 | 0 | 96.7 | 94.7 | 76.8 | 29.2 |
| | DD | 75.0 | 0 | 75.0 | 0 | 90.0 | 0 | 100 | 47.6 | 77.7 | 30.8 |
| | BD | 90.6 | 87.5 | 85.7 | 100 | 0 | 0 | 98.2 | 76.0 | 88.5 | 50 |
| | MT | 84.1 | 36.1 | 66.6 | 0 | 77.9 | 100 | 94.3 | 40.0 | 82.6 | 35.3 |
| | MI | 56.3 | 20.3 | 68.3 | 5.0 | 75.0 | 25.0 | 76.2 | 26.3 | 83.8 | 32.6 |
| | CT | 81.2 | 63.6 | 75.0 | 0 | 0 | 0 | 100 | 95.2 | 98.1 | 70.4 |
| | **Avg** | **80.2** | **39.2** | **77.4** | **39.4** | **52.8** | **23.1** | **92.6** | **60.1** | **84.3** | **39.4** |
| FOL | EI | 100 | 100 | 95.2 | 100 | 100 | 83.3 | 100 | 100 | 83.3 | 100 |
| | UI | 100 | 71.4 | 94.7 | 90.5 | 51.3 | 100 | 90.9 | 100 | 83.3 | 100 |
| | MP | 97.6 | 78.9 | 76.5 | 100 | 79.6 | 90.5 | 82.8 | 95.4 | 91.3 | 79.4 |
| | HS | 100 | 48.8 | 100 | 90.9 | 100 | 76.9 | 98.2 | 76.0 | 97.9 | 59.4 |
| | DS | 75.0 | 25.0 | 75.0 | 0 | 96.6 | 85.7 | 100 | 57.1 | 84.5 | 100 |
| | CD | 78.6 | 80.0 | 75.9 | 100 | 85.7 | 0 | 100 | 100 | 76.0 | 40.0 |
| | DD | 77.0 | 50.0 | 75.0 | 0 | 68.4 | 0 | 100 | 37.0 | 88.4 | 40.5 |
| | BD | 83.3 | 100 | 75.0 | 0 | 50.0 | 23.6 | 100 | 74.1 | 94.2 | 60.7 |
| | MT | 95.3 | 48.6 | 68.3 | 5.0 | 75.0 | 0 | 100 | 47.6 | 95.7 | 54.5 |
| | **Avg** | **89.6** | **67.0** | **81.7** | **54.0** | **78.5** | **51.1** | **96.9** | **76.3** | **88.3** | **70.5** |
| NM | DRI | 50.0 | 50.0 | 64.5 | 100 | 50.0 | 0 | 72.2 | 68.2 | 69.2 | 85.7 |
| | DRS | 67.2 | 4.5 | 74.7 | 0 | 75.0 | 0 | 63.0 | 0 | 74.0 | 0 |
| | DRD | 100 | 95.2 | 80.0 | 100 | 83.3 | 100 | 100 | 100 | 80.0 | 100 |
| | DRO | 40.0 | 40.0 | 50.0 | 0 | 50.0 | 0 | 100 | 90.9 | 74.1 | 100 |
| | RE1 | 76.9 | 28.6 | 74.7 | 0 | 75.0 | 0 | 97.7 | 51.3 | 80.3 | 66.6 |
| | RE2 | 100 | 100 | 52.6 | 100 | 50.0 | 0 | 100 | 100 | 55.5 | 100 |
| | RE3 | 72.2 | 68.2 | 79.2 | 93.8 | 54.1 | 100 | 69.2 | 85.7 | 62.1 | 81.8 |
| | RAP | 56.8 | 55.8 | 52.6 | 100 | 50.0 | 0 | 86.8 | 83.3 | 70.2 | 100 |
| | **Avg** | **70.4** | **55.3** | **66.0** | **61.7** | **60.9** | **25.0** | **86.1** | **72.4** | **70.7** | **79.3** |

Table 11: Performance of LLMs in few-shot setting in terms of label-wise accuracy on LogicBench(Eval), where $A(Yes)$ and $A(No)$ denote the accuracy for the $Yes$ and $No$ labels, respectively.