# OpenReview forum: "LogicBench: A Benchmark for Evaluation of Logical Reasoning"
_NeurIPS.cc/2023/Track/Datasets_and_Benchmarks — Submitted to NeurIPS 2023 Datasets and Benchmarks_

### Official Review · Reviewer_dRyV · 2023-07-21
**A Dataset Generated with Well-Structured Logic, but Requiring Improvement**

**Rating:** 5
**Confidence:** 3
**Correctness:** The section on the dataset has been s…

**Strengths:**

This paper is well-structured and easy to follow. The dataset has been developed to overcome the limitations of existing datasets, which include a restricted set of logics. Particularly, it is referred to as the first benchmark for including non-monotonic reasoning that is not directly formalizable in the first-order setting.


**Additional Feedback:**

It is important to consider how prompts are given when testing the model in order to conduct more rigorous experiments. Furthermore, I'm curious about how training the model on the proposed dataset would affect its ability to handle complex logic-based tasks such as mathematics.

**Clarity:**

The description of how the dataset was created in relation to logic is well-written and provides sufficient understanding.

**Documentation:**

The data collection section has been written in a concise manner, ensuring reproducibility. The data can be accessed through the provided link for verification.

**Ethics:**

No.

**Limitations:**

Section 5 briefly addresses future work. While it is not a critical issue, providing more explicit discussions on the limitations and potential improvements would enhance the understanding of the paper.


**Opportunities For Improvement:**

There are some areas for improvement in the experiments, particularly regarding Table 6. It would be beneficial to include comparisons with models trained on existing logic data or other logic-based models. Additionally, for Table 5, it would be helpful to provide not only the separate results for "Yes" and "No," but also the overall results or the ratio of "Yes" and "No" to better understand the data distribution. This would facilitate comparisons with Table 10. Furthermore, it would be beneficial to show the error bars on the experimental results, rather than simply showing the average. For example, in relation to the training process, error bars can demonstrate the results obtained from using different seeds.
There is potential for improvement regarding the dataset generation process. This is because there are aspects where the answers may vary according to common sense. For example, the dataset has a fixed relationship between "usually," "normally," or "typically" and the answer, but there can be variations in the answers.
Also, it would be beneficial to have an analysis regarding the factors that determined the dataset size and the diversity of the dataset. The meaning of "single-task" and "multi-task" in Table 6 is not clear. It would be better to maintain consistency by using "First-order" instead of "First Order" throughout the paper, as seen in Figure 1.


**Relation To Prior Work:**

A clear contribution is evident in this study, as it aims to create a dataset that is robust and encompasses a greater variety of logic types compared to existing datasets.

**Summary And Contributions:**

This paper introduces the LogicBench dataset, which includes propositional, first-order, and non-monotonic logics. Each data sample consists of triplets of (context, question, answer) generated using both the GPT-3 and rule-based methods. To explain further, GPT-3 generates base sentences using a given prompt, which includes examples in the form of (proposition, context, question), logic definitions used in (context, question), and generation constraints. Rules are then applied to these base sentences and their negation to generate (context, question, answer) triplets that align with reasoning patterns. The generated dataset is LogicBench (Aug), and human quality control is employed to create LogicBench (Eval). The paper analyzes the performance of existing models, such as FLAN-T5, Tk-instruct, UnifiedQA, GPT-3, and ChatGPT, on LogicBench (Aug). Furthermore, the study demonstrates that LogicT5, which is trained on LogicBench (Aug) using T5-Large, outperforms T5-Large on LogicBench (Eval). In summary, the main contribution of this work is the creation of a dataset that includes various kinds of logic, surpassing individual existing datasets, and highlighting the importance of the dataset through experimental results.

---

> ### Author Response · Authors · 2023-08-24
>
> Thank you for your comments and suggestions for improving the paper.
>
> --------------------------
> Q1: Comparison with models trained on existing logic data or other logic-based models.
>
> To avoid the discrepancy between different model architectures, we choose to use the same model architecture to demonstrate the effect of logiBench. In our experiments, the sole distinction between T5 and LogicT5 lies in the latter being trained on LogicBench.This observation denotes that incorporating LogicBench benefits logical reasoning capabilities for downstream tasks (e.g., LogiQA).
>
> ----------------------------
> Q2: Update Average and Error bars in Table 5
>
> We updated it in the Appendix F of the paper (L132-134).
>
> ------------------------------------
> Q3: There is potential for improvement regarding the dataset generation process. This is because there are aspects where the answers may vary according to common sense.
>
> Thanks for the comment, this can be interesting for future work. At this point, we do not incorporate this idea because such variation is subjective (even different individual might have a different answer) and thus cause difficulty for the evaluation. Furthermore, we would like to highlight the main aim of LogicBench is to assess the logical reasoning ability of LLMs.
>
> --------------------------------
> Q4: The meaning of "single-task" and "multi-task" in Table 6 is not clear.
>
> We added the explanation to the paper. Specifically, "single-task" denotes fine-tuning and evaluation on one dataset, while "multi-task" indicates fine-tuning on all four datasets combined, with separate evaluations for each dataset.
>
> -----------------------------------
> Q5: "First-order" instead of "First Order"
>
> We updated the paper.
>
> ------------------------------------
> Q7: Experiments on Math datasets
>
> To investigate this point, we fine-tuned T5-large and LogicT5 models on two mathematics datasets, namely, DROP and MathQA. Since LogicT5 is a generative model, we evaluate performance using Rouge-L. Specifically, on the MathQA, LogicT5 achieves 42.58 rouge-L (slightly higher than 41.34 of T5-large). Furthermore, on the DROP, both models achieve similar performance (50.72 for LogicT5 and 50.94 for T5-large). LogicT5 and T5-large achieve similar performance on both datasets. We attribute this to difference in the numerical reasoning and logical reasoning tasks.

---

> > ### Author Response · Authors · 2023-08-28
> >
> > Dear Reviewer,
> >
> > We trust this message finds you well. We are writing to follow up on the rebuttal response we submitted in response to your feedback. We are eager to receive your insights and comments on our response. If there are any areas that you still find unclear or require further clarification, we would be more than happy to provide additional information to ensure a comprehensive understanding of our work.
> >
> > Your feedback is of utmost importance to us, and we greatly appreciate your time and consideration in evaluating our research.
> >
> > Thank you for your attention, and we look forward to hearing from you soon.

---

### Official Review · Reviewer_9HEk · 2023-07-21
**Useful dataset for assessing and improving reasoning capabilities of LMs**

**Rating:** 7
**Confidence:** 3
**Correctness:** Data collection and experiments seem …
**Clarity:** The paper is well written and easy to…

**Strengths:**

- A QA dataset with instances exhibiting different reasoning patterns is, to the best of my knowledge, novel. This is a welcome contribution as it allows to diagnose errors in reasoning patterns for LLMs and track the progress of newer LLMs in reasoning-heavy tasks.
- The results regarding training on LogicBench indicate that the dataset might also be useful for improving the reasoning abilities of LMs
- The paper is well-written and easy to follow.


**Additional Feedback:**

None

**Documentation:**

The dataset comes with a README. However, documentation could be improved by also commenting on availability and maintenance, and ethical and responsible use.

**Ethics:**

I don't see ethical issues with the work.

**Limitations:**

The authors briefly discuss the limitations of their work in Section 5. However, this could be expanded and also should include possible societal impact.

**Opportunities For Improvement:**

I only have minor comments:
- The fine-tuning experiments should be described in more detail. It would be especially important to mention all important hyperparameters, such as optimizer, learning rate, batch size etc.
- It is not fully clear to me why the authors chose to report the accuracy for yes and no separately.
- The abstract states that the LLMs ``do not fare well on LogicBench, 16 achieving just above random accuracy on average (∼ 52%)''. However, the average results in Table 5 are often consistently much higher than 52%. Is there a contradiction?
- The authors could expand their description of the data quality. How exactly was the human-in-the-loop component implemented? Did you assess annotator reliability, e.g. with inter-annotator agreement metrics?

**Relation To Prior Work:**

The discussion of prior work is adequate with a much more thorough treatment in the Appendix.

**Summary And Contributions:**

The paper ``LogicBench: A Benchmark for Evaluation of Logical Reasoning'' introduces a novel QA dataset that systematically tests reasoning patterns of LLMs. These patterns are derived from the three different logic formalisms propositional, first-order and non-monotonic logics. The authors evaluate five widely-used LLMs and find that in some cases they perform only marginally better than the random baseline. Additionally, the authors show that fine-tuning a T5 model on a training portion of the LogicBench results in a model that performs well on the test set of LogicBench and can even improve performance on other logic-based QA datasets.

---

> ### Author Response · Authors · 2023-08-24
>
> Thank you for your comments and suggestions for improving the paper.
>
> ------------------------
> Q1: Further details on Fine-tuning experiments
>
> We have included the details of hyperparameters in Appendix B.1.
>
> ---------------------------
> Q2: Further details on separately reported (Yes/No) accuracies
>
> Here, the label ‘yes’ signifies that the conclusion provided in the question is entailed by context, and the label ‘no’ signifies that the conclusion provided in the question is not entailed by context. It is important to see the model behavior for both cases. Thus, we have reported separate accuracies to address these cases distinctly.
>
> -----------------------------
> Q3: ``do not fare well on LogicBench, achieving just above random accuracy on average (∼ 52%)''. Is there a contradiction?
>
> The abstract's conclusion is derived from the collective average results presented in Table 5. Our intention was to draw a unified conclusion based on the entire data from Table 5.
>
> ---------------------------------------
> Q4: The authors could expand their description of the data quality.
>
> In this paper, we use “human-in-loop” for the authors who conducted a review of the logical formations to ensure they adhered to the intended structure. The authors examined each reasoning pattern for any potential discrepancies, ensuring that they were logically sound and correctly represented the intended relationships between propositions. In addition to the logical formation, they also dedicated considerable effort to eliminating typos and validating the grammar. Section 2.3 describes these details.
>
> -------------------------------------------
> Q5: Update README
>
> Thanks for this comment. We improved the README. Please check (https://anonymous.4open.science/r/LogicBench-EEBB/README.md).

---

> > ### Author Response · Authors · 2023-08-28
> >
> > Dear Reviewer,
> >
> > We trust this message finds you well. We are writing to follow up on the rebuttal response we submitted in response to your feedback. We are eager to receive your insights and comments on our response. If there are any areas that you still find unclear or require further clarification, we would be more than happy to provide additional information to ensure a comprehensive understanding of our work.
> >
> > Your feedback is of utmost importance to us, and we greatly appreciate your time and consideration in evaluating our research.
> >
> > Thank you for your attention, and we look forward to hearing from you soon.

---

> > ### Comment · Reviewer_9HEk · 2023-08-29
> >
> > Thank you for your detailed response and for incorporating my suggestions! I share some of the concerns of the other reviewers, especially regarding the template-based design raised by reviewer oWM8 and thus leave my (positive) evaluation unchanged.

---

### Official Review · Reviewer_p5i9 · 2023-07-21
**LogicBench: A Benchmark for Evaluation of Logical Reasoning**

**Rating:** 6
**Confidence:** 4
**Correctness:** (everything is fine)
**Clarity:** Yes, except that some points need cla…

**Strengths:**

Natural language inference datasets are critical in evaluating natural language understanding systems but are difficult to build. This paper shows a clever use of language model for the generation of such a dataset.
Targeting a specific inference rule in each problem, and covering 25 inference rules from three kinds of logic, is an interesting feature.

**Additional Feedback:**

Concerning variation generation, I would expect a more systematic description of the process. If there is a distinct procedure for each inference rule (instead of a single compact procedure for all rules), then please include them in the Appendix.
For Modus Tollens, for instance, there seem to be four "propositional slots" in a (Context, Question, Answer) triple — three in the Context and one in the Question — so if one considers two pairs of propositions (S1, ¬S1), (S2, ¬S2), one can build 4^4 triples if any proposition can be found in any slot. I believe that you generate less triples by constraining which propositions can appear in which slots, but how exactly, and why?

I think Existential Instantiation and Universal Instantiation could be explained (in particular because these rules are noted with symbols that are not introduced in the paper).

I don't understand l.162-166. Could you be more explicit?
Then you write "we believe that LogicBench(Eval) provides a more reliable evaluation benchmark"; more reliable than what? and what makes you believe this?

L.200, the two single- and multi-task settings are not explained.

L.223 What is "axiom-wise performance"?

L.286, "do not perform well on LogicBench, even though it requires application of only a single inference rule in each data instance". I think this is incorrect or at least misleading; negative cases in fact require to check every inference rule.

In the abstract, l.5 "the crucial skill pertaining to ‘logical reasoning’ has remained underexplored": what is the crucial skill referred to here?

**Documentation:**

(everything is fine)

**Ethics:**

(everything is fine)

**Limitations:**

(everything is fine)

**Opportunities For Improvement:**

I think there are problems with the text in its current form, as some crucial information are lacking or unclear.
First, it is not very clear exactly what is done during the variation generation step (more on this in the Additional Feedback field below).
Second, it is stated that "LogicBench(Aug) is only a synthetically augmented version for training purposes". I expect more precision here.
Third, I find the Related Work section not comprehensive and systematic enough. What are the main features that really differentiate LogicBench from the other datasets mentioned (and other not mentioned, e.g. FraCas, MED, SICK, HANS or CAD)? The paper would benefit a broader discussion of logic and NLI datasets.

In addition, some experiments (typically, "How do LLMs reason step-by-step?") feel a bit too exploratory. I suggest focusing on the most developed parts of the work.

(more minor issues in the Additional Feedback field below)

**Relation To Prior Work:**

As written above, I find the Related Work section not comprehensive and systematic enough. What are the main features that really differentiate LogicBench from the other datasets mentioned (and other not mentioned, e.g. FraCas, MED, SICK, HANS or CAD)? The paper would benefit a broader discussion of logic and NLI datasets.

**Summary And Contributions:**

The paper details a method for generating natural language inference problems (given a context, the goal is to determine whether some proposition is necessarily or usually true) by prompting a generative language model.
They present LogicBench, a dataset generated according to this method and designed for the evaluation of the logical ability of large language models. Each problem is centered around one of 25 inference rules from three kinds of logic (propositional logic, first-order logic, non-monotonic logic).
Experiments on a variety of systems are performed in order to assess their performance.

---

> ### Author Response · Authors · 2023-08-24
>
> Thank you for your comments and suggestions for improving the paper.
>
> ----------------------
> Q1: Clarification on “variation generation”
>
> We would like to clarify that generating variations for each inference rule doesn't involve distinct procedures; the variations primarily differ in terms of the number of slots used in context and question. To illustrate, let's consider Modus tollens where there are two sets of propositions: (S1, ¬S1) and (S2, ¬S2). If we allow any proposition to occupy any slot then we CAN generate 4^4 combinations. However, it's important to note that not all combinations are valid (logically correct) because some combinations will not abide by the inference rule, rendering the logical correctness of that combination incorrect. For instance, the context "If S1, then ￢S1" lacks logical consistency, leading us to eliminate such combinations. Furthermore, certain combinations are essentially mirror images of others  (i.e., (S1, ¬S2) and (¬S2, S1)), offering no substantial diversity in the dataset. By removing these combinations, we refine and restrict the set of triplets to exclude those that undermine the validity of the inference rule and hence also maintain diversity in our dataset. We added this explanation to the paper in (L153-L155).
>
> --------------------------------------
> Q2: Additional Details to Related Work
>
> We further elaborated (L279-281) mentioning differences of LogicBench with existing datasets. Our related work indeed includes datasets for logic and NLI (i.e., LogicNLI and TaxiNLI), furthermore, we added the suggested datasets in the extended related work section (Appendix H in the paper).
>
> ------------------------------------
> Q3: More Details on Existential Instantiation and Universal Instantiation
>
> Thank you for this comment. We added these details in Appendix B.3. This will further improve the readability of this section.
>
> -------------------------------------
> Q4: "we believe that LogicBench(Eval) provides a more reliable evaluation benchmark"; more reliable than what? and what makes you believe this?
>
> LogicBench(Eval) is manually verified and corrected by authors in order to maintain its quality and correctness. Furthermore, this dataset covers diverse inference rules spanning three logic types while existing works cover only a few rules that are limited logic types. Thus, our work serves as a more reliable evaluation benchmark for measuring logical reasoning ability.
>
> ---------------------------------
> Q5: Details about single- and multi-task settings.
>
> We have added this detail in Section 3.1 (L202-204) in the paper. Specifically, "single-task" denotes fine-tuning and evaluation on one dataset, while "multi-task" indicates fine-tuning on all four datasets combined, with separate evaluations for each dataset.
>
> ----------------------------------
> Q6: L.223 What is "axiom-wise performance"?
>
> Thanks for highlighting this point. This is a typo. We have corrected this in the revised version. Here, we indicate performance corresponding to each inference rule.
>
> -----------------------------------------------
> Q7: Clarification on L.286, "do not perform well on LogicBench, even though it requires application of only a single inference rule in each data instance".
>
> Thanks for pointing this out. Here, we wanted to indicate that the questions with a ‘Yes’ label (i.e., positive instances where the question is entail by context) require the application of only a single inference rule in each data instance. We corrected it and will further clarify this in the paper.

---

> > ### Author Response · Authors · 2023-08-28
> >
> > Dear Reviewer,
> >
> > We trust this message finds you well. We are writing to follow up on the rebuttal response we submitted in response to your feedback. We are eager to receive your insights and comments on our response. If there are any areas that you still find unclear or require further clarification, we would be more than happy to provide additional information to ensure a comprehensive understanding of our work.
> >
> > Your feedback is of utmost importance to us, and we greatly appreciate your time and consideration in evaluating our research.
> >
> > Thank you for your attention, and we look forward to hearing from you soon.

---

> > ### Comment · Reviewer_p5i9 · 2023-08-29
> >
> > I have one last question/remark about Q1.
> > True, not all combinations satisfies the inference rule, but is this a reason to reject them? Why not using them as negative instances?
> > (As for "If S1, then ¬S1", I would not say that this proposition lack consistency. What lacks consistency, however, is S1, given "If S1, then ¬S1". This proposition simply entails ¬S1 in classical logic.)

---

> > > ### Author Response · Authors · 2023-08-30
> > >
> > > You are right that “If S1 then ~S1” is not logically inconsistent. In fact, logically it entails ~S1. But when we replaced S1 with a natural language statement such as "I am hungry", it would become “If I am hungry then I am not hungry” which did not seem intuitive to us; hence we did not consider such cases. However, we believe that “If S1 then ~S1” and “S1” are logically inconsistent. In that case, logically everything will be entailed by "If S1 then ~S1 " and "S1"; which again did not seem intuitive. Please let us know if you have further concerns or need justification regarding this point.

---

### Official Review · Reviewer_ywMi · 2023-07-21
**Brief review for LogicBench: A Benchmark for Evaluation of Logical Reasoning**

**Rating:** 7
**Confidence:** 3
**Correctness:** Yes, the paper seems correct.
**Clarity:** Yes, overall the paper is well-writte…

**Strengths:**

A novel approach to study non-monotonic reasoning, as well as various inference rules.

LLMs trained using proposed data exhibit a better understanding of logical reasoning leading to performance improvements.

Good amount of comprehensive experiments was provided.

This area is important and clearly is relevant to benchmarking in general.

**Additional Feedback:**

None

**Documentation:**

Yes, the documentation is sufficient.

**Ethics:**

No ethical concerns.

**Limitations:**

Yes, the authors addressed the limitations in the presented work.

**Opportunities For Improvement:**

More well-known and up-to-date LLMs could be added to the experiments e.g. Bard, GPT-4

**Relation To Prior Work:**

Yes, the cover of paper's prior work is sufficient.

**Summary And Contributions:**

The authors proposed LogicBench, a systematically created question-answering dataset for the evaluation of logical reasoning ability, which consists of 25 distinct inference rules and reasoning patterns. Two versions of dataset was released: LogicBench(Eval) for evaluating LLMs and LogicBench(Aug) for training purposes. The authors demonstrated that LLMs trained using LogicBench(Aug) showcase an improved understanding of logical reasoning, resulting in a better performance on existing logic datasets.

---

> ### Author Response · Authors · 2023-08-24
>
> Thank you for your comments and suggestions for improving the paper.
>
> --------------------
> Experiments on Bard, and GPT-4
>
> Thank you for this suggestion. Experiments on Bard and GPT-4 are presented in Appendix G. We can observe that even Bard and GPT-4 achieve an average accuracy of just 51.2% and 62.8% on LogicBench(Eval), indicating their limited ability to perform logical reasoning. This further highlights the scope of improvement in the logical reasoning ability of these models.

---

> > ### Author Response · Authors · 2023-08-28
> >
> > Dear Reviewer,
> >
> > We trust this message finds you well. We are writing to follow up on the rebuttal response we submitted in response to your feedback. We are eager to receive your insights and comments on our response. If there are any areas that you still find unclear or require further clarification, we would be more than happy to provide additional information to ensure a comprehensive understanding of our work.
> >
> > Your feedback is of utmost importance to us, and we greatly appreciate your time and consideration in evaluating our research.
> >
> > Thank you for your attention, and we look forward to hearing from you soon.

---

### Official Review · Reviewer_oWM8 · 2023-07-22
**An interesting proposal, but with flawed task and data formulations**

**Rating:** 4
**Confidence:** 4
**Correctness:** 1. Datasets

**Strengths:**

1. Problem Proposal: I appreciate the efforts authors paid to arouse public attention on LLMs' logical inference. While LLMs' have been demonstrate strong grasping of human languages, the systematic reasoning hasn't been fully evaluated.
2. Interesting Taxonomy: LogicBench is an endeavor to systematically connect conventional taxonomy of logics to LLMs' reasoning beyond the prior common practice in NLP community. Such theoretical guide is valuable.

**Additional Feedback:**

See above limitation discussion.

**Clarity:**

The paper is easy to follow, but I think it should provide more thorough analysis on the data quality (the current Section 2.3 lacks details) and diversity. Also, the evaluation on LogicT5 is insufficient (i.e., Table 6).

**Documentation:**

The authors present enough details.

**Limitations:**

See opportunities for improvement.

**Opportunities For Improvement:**

1. Task formulation: A major issue with this work is the task formulation. When it comes to LLMs' zero-shot reasoning, the Chain-of-Thought (CoT) [1] prompting has become a de facto practice for meaningful reasoning. However, in this work LogicBench still only adopts the binary classification, which does not reflect the reasoning process of LLMs. Despite authors present post-hoc explanatory analysis in CoT-similar style in the Section 3.3 analysis and appendix, it is not for the major quantitative evaluation part in this work. I think a systematic look into CoT evaluation results is necessary for this work..
2. Template-based data creation: Another potential problem lies in the data creation, especially when it comes to fine-tuning for LogicT5. I check the data provided in the anonymous GitHub and find them to be monotonous in template formats (i.e., for samples in the same inference rule). It could be problematic, especially as the authors report LogicT5's "~97% of accuracy on LogicBench(Eval)" after they trained T5-large on LogicBench(Aug) --- which means the evaluation could be easily hacked even using a relatively small language model. I think authors should create more diverse templates and formats to ensure the level of challenge in LogicBench.

**Relation To Prior Work:**

The work clearly discusses its difference to prior works.

**Summary And Contributions:**

The paper introduces an interesting view sight on evaluating LLMs' logical reasoning capabilities regarding three different logics: propositional, first-order, and non-monotonic. Authors create a large new logical dataset based on templates via ChatGPT for both training and evaluation. They also train a LogicT5 to show that their data can help LLMs to improve logical inference ability.

---

> ### Author Response · Authors · 2023-08-24
>
> Thank you for your comments and suggestions for improving the paper.
>
> ------------------
> Q1: Task formulation and experiments with CoT
>
> We agree that CoT is the standard practice for meaningful reasoning with LLMs. Thus, we have included the results of GPT-3 and ChatGPT using CoT prompting in Appendix E. In the same line, we further analyzed instances corresponding to inference rules on which the model’s performance is low. Specifically, we evaluate whether the predicted logical reasoning steps correlate with the prediction performance. This experiment resulted in several interesting findings, such as revealing that LLMs have difficulty in establishing logical relationships between sentences and they fail to capture the nuanced meanings of words such as “usually”', “normally” and “typically”. We have detailed the experimental setup and other results in Section 3.3 and Appendix E.
>
> ----------------------
> Q2: Template-based data creation and fine-tuning
>
> The emergence of the prompt paradigm has shifted the assessment of LLMs, favoring zero-shot and few-shot evaluations over traditional supervised fine-tuning (SFT). Reflecting this trend, LogicBench has been designed to serve as an evaluation benchmark specifically targeting the logical reasoning capabilities of LLMs. Here, we evaluate various LLMs in zero-shot (Section 3) and few-shot (Appendix F) settings. Our findings reveal that these models struggle with simple inference rules spanning diverse logic types. It is important to note that LLMs are not trained on LogicBench, thereby preventing them from relying on potential dataset-specific patterns. Hence, our reported LLM performance indeed reflects their logical reasoning capabilities. Additionally, our paper delves into fine-tuning with LogicBench, presented as an additional study, showcasing the impact of fine-tuning and its usefulness towards downstream tasks involving logical reasoning (Section 3.3).
>
> -------------------------------
> Q3: Qualitative Analysis and Evaluation of LogicT5
>
> Here, we use a template-based approach to create data instances, ensuring that generated sentences strictly follow the structure of inference rules and maintain logical correctness. Additionally, our dataset is diverse in terms of incorporating 25 different inference rules spanning distinct three logic types. In Appendix B, we have included qualitative analysis where we plot word cloud showing diverse ontology terms along with logic terms in our dataset.
>
> We would also like to highlight that the main objective of the work is to provide an evaluation benchmark to asses the logical reasoning abilities of LLMs. However, as an additional study, we demonstrate other possible benefits of the proposed data such as fine-tuning on LogicBench(Aug) to help other downstream tasks involving logical reasoning (Section 3.3).

---

> > ### Author Response · Authors · 2023-08-28
> >
> > Dear Reviewer,
> >
> > We trust this message finds you well. We are writing to follow up on the rebuttal response we submitted in response to your feedback. We are eager to receive your insights and comments on our response. If there are any areas that you still find unclear or require further clarification, we would be more than happy to provide additional information to ensure a comprehensive understanding of our work.
> >
> > Your feedback is of utmost importance to us, and we greatly appreciate your time and consideration in evaluating our research.
> >
> > Thank you for your attention, and we look forward to hearing from you soon.

---

> > ### Comment · Reviewer_oWM8 · 2023-08-31
> >
> > Thanks for authors' responses.
> >
> > I think the complementary CoT experiments are useful. The reported performance degradation of LLMs (i.e., GPT-3, ChatGPT) in this setting indicates that the original main results could be dubious, since in common practice CoT should *improve* LLM reasoning. In fact, I believe this should be the main setting of this benchmark, and more LLMs with CoT abilities should be tested to make it a convincing benchmark.
> >
> > For template-based data creation, I still hold that it is a significant problem, especially as the LogicT5 shows that small LMs' LogicBench performance could be outrageously high after simple fine-tuning. I think the authors should at least try to reformulate the templated-based inputs into human-like natural languages to test LMs' real abilities in understanding and reasoning human questions that relate to logical reasoning.
> >
> > In conclusion, I think this paper still requires significant rewriting and improvement before publishing. I will hold my original scores. Thanks again for authors' efforts.

---

### Author Response · Authors · 2023-08-24
**General Response**

We thank the reviewers for their insightful comments. We are encouraged that the reviewers find our dataset on the evaluation of the logical reasoning capabilities of LLMs to be “novel”, “valuable”, “systematic”, and “diverse”. We are also pleased that reviewers appreciate our efforts to create a “dataset of interesting taxonomy of logic to LLMs' reasoning beyond the prior common practice in the NLP community”. Furthermore, reviewers mention the importance of our dataset in diagnosing errors in reasoning patterns for LLMs and tracking the progress of newer LLMs in reasoning-heavy tasks. Moreover, we are pleased that reviewers appreciate our dataset as it is the first benchmark to include non-monotonic and various other inference rules. We would like to thank the reviewers for providing comments and feedback. We updated the paper and supplementary material accordingly and submitted a revised version.

---

### Comment · Area_Chair_126n · 2023-08-29
**Please acknowledge author's responses**

Dear reviewers,

Thank you again for all your hard work.

The author's responses came in a few days ago and it would be helpful if reviewers could

take time to read the author's rebuttals,
acknowledge them by responding,
lastly, change scores if needed.
This paper has mixed scores and having further discussions with authors could be very helpful for the final evaluation of this paper.

Also sharing the reviewer guidelines for D&B track just in case as there are some distinct differences with the regular track: https://neurips.cc/Conferences/2023/DatasetsAndBenchmarks/ReviewGuidelines

Best, AC

---

### Decision · Program_Chairs · 2023-09-22

**Decision:**

Reject

**Comment:**

Authors claim that previous datasets are limited to only a couple of logical axioms of propositional and first-order logic. In order to rightfully assess LLM’s logical capabilities,  the authors present LogicBench, where each problem is generated (using ChatGPT and templates) using 25 inference rules that provide three kinds of logic (propositional, first-order, and non-monotonic).

Including myself, the reviewers seem to share the importance of measuring the logical capabilities of LLMs and creating benchmarks to do so. However, there were some concerns about this dataset: (1) the generation process is too monotonous (i.e. as T5 results hint that the dataset is hackable), (2) Some comparison with previous datasets (not just in writing) could be helpful.

Lastly, there were also requests to make the generation process and details a bit clearer from reviewers p5i9, 9HEk, and dRyV. Individually, these could look minor, however, combined with other technical suggestions such as putting CoT as the main story, or diversifying the generation process, I think this paper can greatly improve if the authors reflect the reviewer’s recommendation in their next submission.